# Engineered immunogens to elicit antibodies against conserved coronavirus epitopes

A. Brenda Kapingidza[1,2,10], Daniel J. Marston[1,2,10], Caitlin Harris [1,2], Daniel Wrapp[1,2], Kaitlyn Winters[1,2], Dieter Mielke [3,4], Lu Xiaozhi[1,2], Qi Yin[1,2], Andrew Foulger[1,2], Rob Parks[1,2], Maggie Barr[1,2], Amanda Newman[1,2], Alexandra Schäfer [5], Amanda Eaton [3], Justine Mae Flores[1,2], Austin Harner[1,2], Nicholas J. Catanzaro Jr.[5], Michael L. Mallory[5], Melissa D. Mattocks[5], Christopher Beverly[1,2], Brianna Rhodes[1,2], Katayoun Mansouri[1], Elizabeth Van Itallie[1,2], Pranay Vure[1,2], Brooke Dunn[3], Taylor Keyes[3], Sherry Stanfield-Oakley[3], Christopher W. Woods [1,2,6], Elizabeth A. Petzold[6], Emmanuel B. Walter[1,7], Kevin Wiehe[1,2], Robert J. Edwards [1,2], David C. Montefiori [3], Guido Ferrari[1,3,4,8], Ralph Baric [5], Derek W. Cain[1,2], Kevin O. Saunders [1,3,8,9], Barton F. Haynes [1,2,9] & Mihai L. Azoitei [1,2] ✉

Immune responses to SARS-CoV-2 primarily target the receptor binding domain of the spike protein, which continually mutates to escape acquired immunity. Other regions in the spike S2 subunit, such as the stem helix and the segment encompassing residues 815-823 adjacent to the fusion peptide, are highly conserved across sarbecoviruses and are recognized by broadly reactive antibodies, providing hope that vaccines targeting these epitopes could offer protection against both current and emergent viruses. Here we employ computational modeling to design scaffolded immunogens that display the spike 815-823 peptide and the stem helix epitopes without the distracting and immunodominant receptor binding domain. These engineered proteins bind with high affinity and specificity to the mature and germline versions of previously identified broadly protective human antibodies. Epitope scaffolds interact with both sera and isolated monoclonal antibodies with broadly reactivity from individuals with pre-existing SARS-CoV-2 immunity. When used as immunogens, epitope scaffolds elicit sera with broad betacoronavirus reactivity and protect as "boosts" against live virus challenge in mice, illustrating their potential as components of a future pancoronavirus vaccine.

The majority of immune responses elicited against SARS-CoV-2, by either natural infection, vaccination or a combination of both, are focused on the receptor binding domain (RBD)[1–4]. However, deep scanning mutagenesis studies[5] and the emergence of novel variants that evade pre-existing immunity have revealed the plastic nature of the RBD[2,6], suggesting that next-generation coronavirus vaccines designed to protect against both current and emergent CoVs will need to induce antibodies against CoV spike regions beyond the RBD.

Two regions in the S2 subunit of the SARS-CoV-2 spike, the segment encompassing residues 815-823 adjacent to the fusion peptide and the stem helix domain (residues 1145-1156), show high conservation across diverse coronaviruses. Isolated antibodies that target these sites cross-react with both alpha and beta human coronaviruses, and

protect by either neutralization, Fc receptor mediated mechanisms or both[7–10]. For example, antibodies DH1058, DH1294, VN01H1, Cov44-79, and Cov44-62, bind to an epitope located around SARS-CoV-2 spike residues 815-823 and recognize all seven human CoVs[7,8,11,12]. Antibodies against the stem helix region, such as S2P6 and CC40.8[9,10,13,14], bound spike glycoproteins representative of all sarbecovirus clades, and protected against viral challenge by inhibiting S-mediated membrane fusion. Despite their attractiveness as vaccine targets, humoral responses against the stem helix or the spike 815-823 epitopes are not robustly induced by existing vaccines or by natural infection[7–10]. This is likely due to a combination of factors, including the occlusion of these epitopes on the pre-fusion spike, requiring ACE2 binding for exposure[8], as well as the presence of the prominently displayed, immunodominant RBD domain. Given their ability to recognize diverse coronaviruses, immunogens that induce strong responses against the spike 815-823 region and the stem helix could be a key component of a future "pan-coronavirus" vaccine offering broad protection against both currently circulating and emergent coronaviruses.

For the design of immunogens that expose occluded epitopes in novel molecular contexts, we and others previously described the development of "epitope scaffold" proteins[15–17]. These molecules are engineered by transplanting the structure of the antibody-bound epitopes from viral molecules onto unrelated protein scaffolds. Unlike peptide-based immunogens that can sample diverse epitope conformations, epitope scaffolds are meant to present only the antibody-bound conformation of the target epitope on their surface, which typically leads to higher antibody affinity and the elicitation of antibodies specific for the structure of the target epitope[17,18]. Here, we applied epitope grafting to design stem helix or spike 815-823 epitope scaffolds that strongly interact with broadly cross-reactive antibodies against these regions of spike. These antigens were used to isolate diverse antibodies with broad reactivity against these conserved epitopes from subjects with pre-existing SARS-CoV-2 immunity. When used as immunogens in mice, epitope scaffolds elicited sera with broad betacoronavirus reactivity. In a viral challenge model, boosting responses induced by spike mRNA with stem helix epitope scaffolds offered protection against a pre-emergent coronavirus, thus illustrating the potential of the immunogens engineered here as next-generation vaccine candidates to preferentially boost antibodies that may protect against existing and emerging coronaviruses.

## Results

### Design of epitope scaffolds that bind broadly cross-reactive antibodies against the spike 815-823 epitope

The region containing residues 815-823 of the SARS-CoV-2 spike, thereafter referred to as spike[815–823], was initially thought to be part of the fusion peptide domain. However, recent structural analysis of the post-fusion spike in a lipid environment revealed that the actual fusion peptide is located upstream of this region (residues 867-909). While the functional role of the spike[815–823] region remains to be determined, multiple broadly protective antibodies, such as DH1058, DH1294, VN01H1, Cov44-79, and Cov44-62 target this site, with key contacts made with virus residues R815, E819 and F823 (Fig. 1a, Supplementary Fig. 1)[7,8,11,12]. These residues are occluded in the structure of the pre-fusion SARS-CoV-2 spike, likely limiting their immune recognition (Fig. 1a). To improve the accessibility and immune recognition of this epitope, we developed spike[815–823] epitope scaffolds (ES) using "side chain" grafting computational methods that we and others previously described (Supplementary Fig. 2)[15–17]. A large library (~10,000) of small monomeric scaffolding proteins was queried computationally to identify proteins with exposed backbone regions that closely matched (<0.5 Å backbone RMSD) the structure of the spike[815–823] epitope (Supplementary Fig. 2). In regions with high structural mimicry to the DH1058-bound epitope, the epitope sequence replaced that of the

parent scaffold and additional mutations were introduced to accommodate the grafted sequence (Supplementary Fig. 2).

Fifteen engineered spike[815–823] epitope scaffolds were expressed recombinantly (Fig. 1b for representative designs), and six of them produced soluble and stable proteins that bound DH1058 mAb by ELISA (Supplementary Fig. 3, Supplementary Dataset 1). Equilibrium dissociation constants for mAbs DH1058 and DH1294 were determined by surface plasmon resonance (SPR) for the designs that showed tight binding by ELISA (Fig. 1c; Supplementary Fig. 4). FP-2, FP-10, and FP-15 bound to DH1058 with picomolar affinities and to DH1294 with $K_D$s of 1221 nM, 0.1 nM and 219 nM respectively. These values compare favorably with measured affinities of the full-length WA-2 spike to DH1058 and DH1294 mAbs (1.4 nM and 25 nM respectively). To probe the breadth of their recognition, the binding of spike[815–823] epitope scaffolds was measured against antibodies VN01H1, C77G12, VP12E7, COV44-79, and COV44-62[7,8], which were not explicitly considered at the design stage (Fig. 1d; Supplementary Fig. 5a). Five designs bound at least 6 out of the 7 antibodies tested, with FP-12 and FP-15 binding all of them tighter than spike. In addition to the mature antibodies, binding was also measured to their inferred unmutated common ancestors (UCAs) or germline-reverted forms (iGLs). Spike only showed measurable binding to the DH1058 UCA, while FP-12 and FP-2 were bound by DH1058 UCA, DH1294 UCA and VP12E7 iGL (Fig. 1d; Supplementary Fig. 5b). Taken together, these data show that the engineered FP epitope scaffolds have broad recognition of genetically diverse FP antibodies and their precursors.

DH1058 mAb bound three spike[815–823] epitope scaffolds with affinities 10-fold higher than that of spike. To ensure that these affinity gains were primarily mediated by the grafted epitope residues and not by contacts between the scaffold and the antibody that are not present in the native DH1058-epitope complex, interface hotspot residues corresponding to SARS-CoV-2 spike amino acids R815, E819, and F823 were mutated to alanine in designs FP-2, FP-10, and FP-15. Each of the three alanine substitutions eliminated DH1058 mAb binding to a synthetic peptide encompassing the spike[815–823] epitope, and to designs FP-2 and FP-10, demonstrating that the grafted epitope is the major site of antibody interaction (Fig. 1e). Two of the three substitutions similarly eliminated DH1058 binding to FP-15, with the F823A equivalent retaining some low-level antibody binding (Fig. 1e).

To further confirm binding specificity and to validate the computational design process, a high resolution (2.2 Å) crystal structure of FP-15 in complex with DH1058 mAb was solved (Fig. 1f, Supplementary Table 1). The experimentally determined structure was in close agreement with that of the computationally generated antibody-ES complex, both overall and over the epitope scaffold only (backbone RMSD < 1 Å). The epitope conformation engaged by DH1058 on FP-15 was essentially identical to that induced by the antibody on the spike[815–823] peptide (backbone RMSD = 0.3 Å)[11], confirming that the antibody interacts with the epitope presented by the ES in the same manner it binds to its natural target.

### Design of epitope scaffolds to engage S2P6-like antibodies against the stem helix region of the CoV spike

Next, we engineered epitope scaffolds to engage antibodies against the stem helix region of the CoV spike. Antibodies such as S2P6, DH1057.1, and CC40.8[9,10,19] bind to an epitope located between residues 1144 and 1158, which is almost completely occluded by the trimerization interface in the pre-fusion spike structure (Fig. 2a)[9,10,19]. While both CC40.8 and S2P6 target the same spike region and make critical contacts with residues Phe1148 and Phe1156, CC40.8 makes additional interactions with Leu1145 and induces a more extended epitope conformation upon binding (Fig. 2a)[10]. Because the CC40.8 epitope is longer and includes the S2P6 epitope, we first attempted to engineer epitope scaffolds that display the antibody-bound conformation of the CC40.8 epitope, with the expectation that these

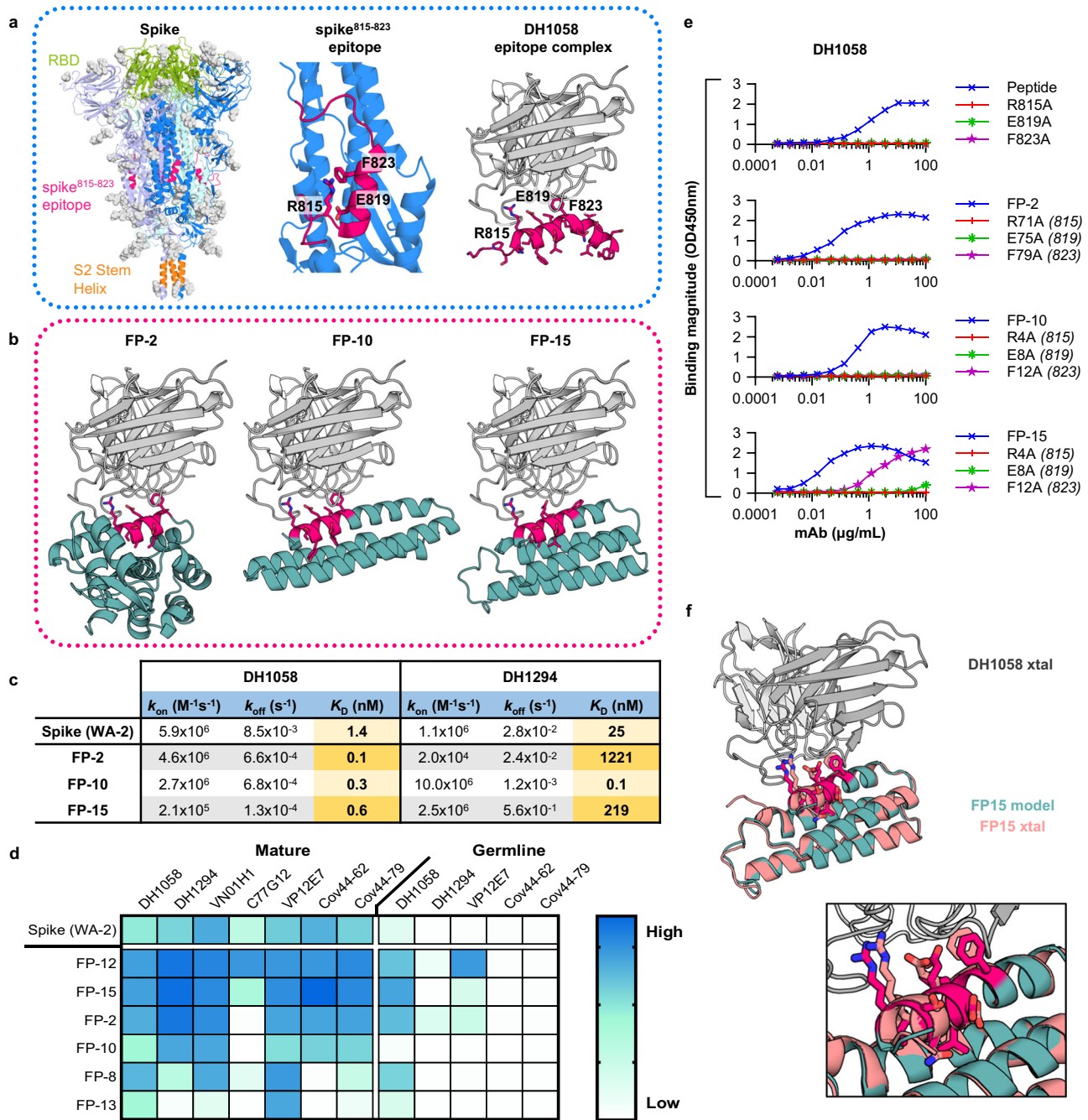

**Fig. 1 | Design of epitope scaffolds that bind to broadly cross-reactive antibodies against the spike[815–823] peptide. a** *Left*: Structure of the pre-fusion spike trimer (individual monomers: *blue, violet and pale green*; glycans: *gray*; PDBid:6xr8), with the Receptor Binding Domain (RBD) (*green*), spike[815–823] peptide (*red*) and stem helix (*orange*) domains highlighted. *Middle*: Zoom of spike monomer with spike[815–823] peptide highlighted (*red*) showing key residues engaged by antibodies. *Right*: Structure of spike[815–823] peptide (*red*) bound to DH1058 mAb (*gray*) (PDBid:7tow). **b** Computational models of DH1058 (*gray*) bound to ESs FP-2, FP-10 and FP-15 (*green*) with the grafted epitope shown in *red sticks*. **c** Binding affinities of ESs and spike to DH1058 and DH1294 mAbs as determined by SPR. **d** Binding of spike and ES to diverse spike[815–823]-targeting mAbs and their inferred precursors. **e** ELISA binding of DH1058 mAb to synthetic spike[815–823] peptides and representative ESs that contain alanine mutations at epitope residues critical for antibody recognition. Numbers in brackets indicate the equivalent position of the mutated epitope residues on spike. **f** Crystal structure of DH1058 mAb (*gray*) in complex with FP-15 (*salmon*) overlaid with the computational model of the ESs (*green*). Epitope residues are shown in sticks (model: *red*; crystal structure: *salmon*). Source data are provided as a Source Data file.

molecules would also be capable of binding to S2P6-class antibodies. However, the "side chain" grafting protocol used for FP epitope scaffolds above, did not identify any candidate scaffolds with exposed backbones similar to that of the CC40.8 epitope, likely due to the unusual conformation of this epitope that adopts a distorted alpha-helical shape near the N-terminus (Fig. 2a). In contrast, multiple candidate epitope scaffolds that displayed the S2P6-bound conformation of the epitope segment 1148-1156 were successfully designed

computationally using side chain grafting. This was possible because the S2P6 epitope adopts a canonical alpha-helical structure, with most of the antibody contacting residues contained within three helical turns (Fig. 2a).

Fifteen S2P6-targeted designs, named S2hlx-1 to 15, were chosen for recombinant production in *E. coli* (Fig. 2a for representative designs; Supplementary Dataset 1) and seven yielded soluble protein. In an initial ELISA screen, all seven designs bound both S2P6 and

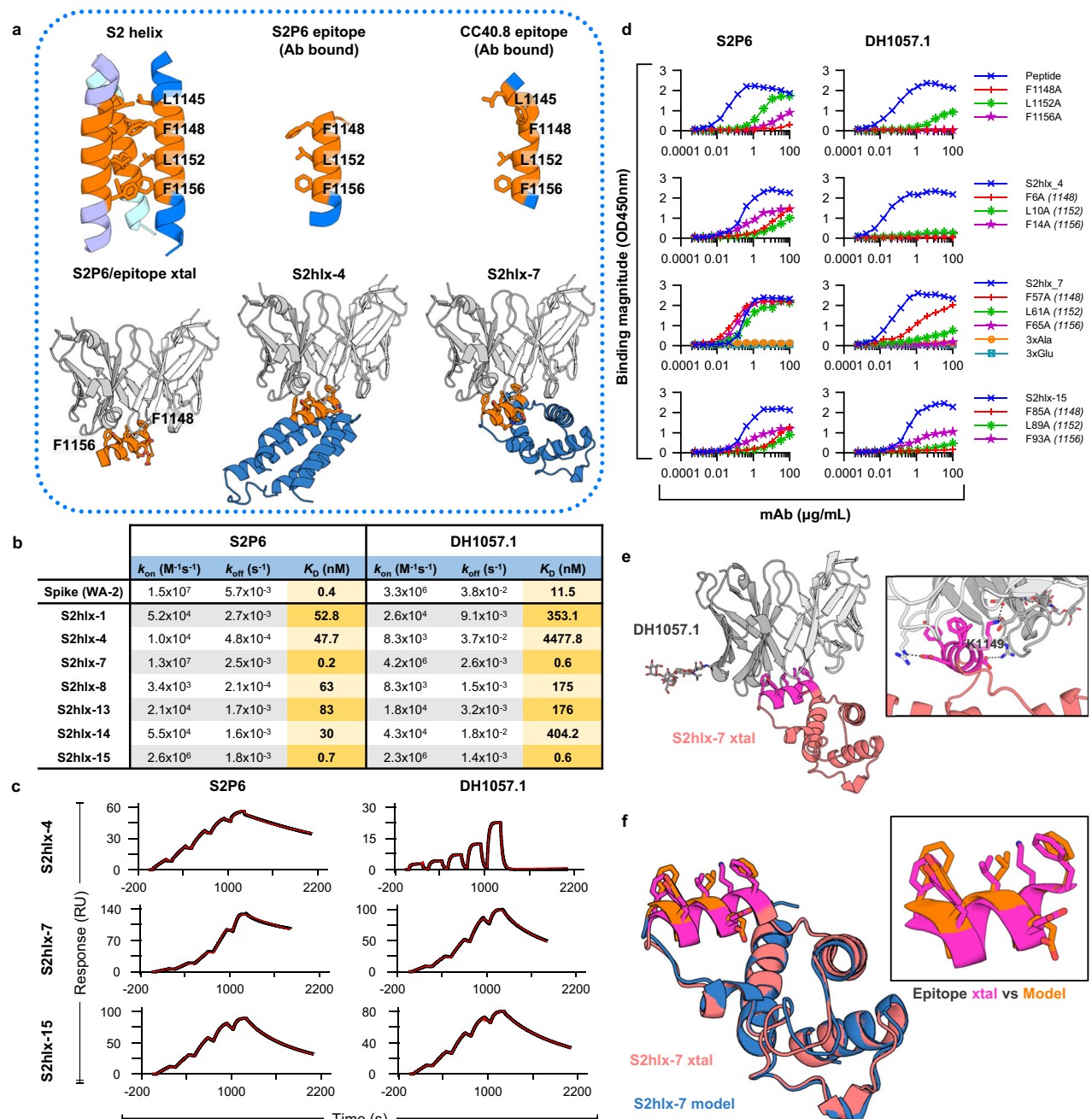

**Fig. 2 | Design of epitope scaffolds that bind to S2P6-like antibodies against the stem helix. a** The conformation of the stem helix epitope (*orange*) on the pre-fusion trimer, bound by S2P6 and CC40.8mAbs (PDBids:7rnj and 7sjs), and displayed on representative epitope scaffolds S2hlx-4 and -7. Key epitope residues are labeled and shown in sticks. **b** Binding affinities of spike and S2hlx ESs to S2P6 and DH1057.1 mAbs. **c** Representative single-cycle binding SPR traces used to determine the dissociation constants in (**b**). **d** The effect on S2P6 and DH1057 mAbs binding to epitope mutations in synthesized stem helix peptides (1140-1164) and representative S2-hlx ESs. **e** Crystal structure (xtal) of the DH1057.1 mAb (*gray*) bound to ES S2hlx-7 (*orange*) with the epitope in *pink sticks*. Inset shows details of the epitope/mAb interaction, with epitope residue K1149 highlighted. **f** Comparison of the S2hlx-7 conformation in the computational model (*pink*) and the crystal structure (*blue*), both overall and in the epitope region. Source data are provided as a Source Data file.

DH1057.1 mAbs comparable to or better than a synthesized SARS-CoV-2 stem helix peptide (residues 1147-1161). As expected, neither this peptide nor the epitope scaffolds bound CC40.8 mAb due to the lack of epitope residue L1145 (Supplementary Fig. 6). Stem helix epitope scaffolds had measured equilibrium dissociation constants between 0.2 nM and 83 nM for S2P6 mAb and between 0.6 nM and 4.5uM for DH1057.1 mAb. In the same assay, the equilibrium dissociation constants of spike for S2P6 and DH1057.1 were 0.4 nM and 11.5 nM, respectively (Fig. 2b, c; Supplementary Figs. 7 and 8). The designs with the highest affinities, S2hlx-7 and S2hlx-15, had similar binding of S2P6

to that of spike and bound DH1057.1 >10-fold better, thus demonstrating the successful transplantation of the target epitope onto the scaffolds.

The binding specificity of S2hlx-4, S2hlx-7 and S2hlx-15 was confirmed by mutating epitope residues equivalent to spike Phe1148, Leu1152 and Phe1156. Alanine mutations limited the binding to both S2P6 and DH1057.1 mAbs in S2hlx-4 and -15 to a similar extent to what as observed for synthesized stem helix peptides (Fig. 2d)[9]. For S2hlx-7, the individual epitope mutations greatly decreased DH1057.1 mAb interactions but had almost no effect on S2P6 mAb recognition.

However, an S2hlx-7 variant that contained alanine substitutions at all three epitope positions completely abrogated binding, thus confirming that antibody interactions were mediated by the grafted epitope residues. In addition, none of the parent scaffolds these designs were based on showed binding to S2P6 or DH1057.1, further demonstrating epitope specificity (Supplementary Fig. 9).

High resolution (1.9 Å) structural determination by x-ray crystallography of the S2hlx-7/DH1057.1 complex revealed for the first time the structure and interaction mode of the DH1057.1 antibody with the stem helix epitope. DH1057.1 engaged the epitope at the same angle as S2P6 mAb and similarly interacted with residues Phe1148, Leu1152, and Phe1156 (SARS-CoV-2 spike numbering) (Fig. 2e), as predicted by the alanine mutagenesis (Fig. 2d). Unlike S2P6, DH1057.1 interacts with residue K1149 through hydrogen bonds mediated by heavy chain residues Asp95 and Asp100C (Fig. 2e). The structure of S2hlx-7 in the complex was closely aligned with that of the computational model both overall (backbone RMSD = 0.5 Å) and, most importantly, over the grafted epitope region (backbone RMSD = 0.25 Å) (Fig. 2f). The epitope conformation displayed on S2hlx-7 closely matched that of the antibody-bound S2P6 epitope and that of the corresponding region on the prefusion spike (backbone RMSD < 0.2 Å, Fig. 2f). These data validated the computational design of antigens that have high affinity and specificity for S2P6-like mAbs by mimicking the conformation of their epitope on native spikes.

## Design of epitope scaffolds that engage stem helix antibodies with different specificities

Given the failure of side chain grafting to produce epitope scaffolds that engage the CC40.8 mAb, we employed an alternative design technique, termed backbone grafting. Here, rather than transplanting only the epitope-specific residues onto a preexisting scaffold backbone which is otherwise unchanged, an entire region of the parent scaffold backbone is replaced with the backbone of the secondary structure containing the desired epitope. This allows the grafting of more complex epitopes that have no existing structural match, such as the CC40.8 S2-helix epitope. Candidate scaffolds were identified by aligning the N- and C-termini of the stem helix epitope in its CC40.8-bound conformation with the N- and C-termini of possible insertion sites on candidate scaffolds (Fig. 3a, b). For scaffolds where the backbone RMSD of this alignment was below 0.75 Å, the epitope replaced the overlapping backbone of the parent scaffold, and additional mutations were introduced to integrate the epitope into the scaffold and to ensure productive interactions with CC40.8 mAb (Fig. 3a). The structures of candidate designs were predicted using Alphafold2[20], and additional mutations were introduced to ensure proper epitope conformation. Final designs contained between 16 to 32 mutations relative to the parent scaffolds, and the grafted epitope was 3.1 to 6.7 Å away by RMSD from the backbone it replaced.

Six epitope scaffolds, named S2hlx-Ex1 to 6 (Fig. 3b for representative models; Supplementary Dataset 1) were tested for bacterial expression and analyzed for binding to S2P6 mAb in an initial screen. While four of the six designs, S2hlx-Ex2, -Ex3, -Ex4 and -Ex6, bound to S2P6, three of them those had poor expression (S2hlx-Ex2, -Ex4 and -Ex6) (Supplementary Fig. 10). To improve expression, we tested different protein expression tags and evaluated structural homologs of the parent scaffolds as alternative design templates with limited success (Supplementary Figs. 11, 12). Next, we optimized the sequence of the S2hlx-Ex designs using ProteinMPNN[21], a recently described deep learning algorithm that takes a protein structure as input and generates amino acid sequences that are predicted to generate the same fold. All four backbone grafting designs that bound S2P6, S2hlx-Ex2, -Ex3, Ex4, and -Ex6, were optimized with this approach, and sequences that expressed with high yield (40 mg/L of culture on average, Supplementary Fig. 13) were identified for all of them. For a given structure, ProteinMPNN generated designs that had highly different

sequences outside the epitope (Fig. 3b). For example, S2hlx-Ex2 based designs -Ex15, -Ex17, -Ex19, -Ex20 had between 42% and 55% amino acid sequence identity with the S2hlx-Ex2 template (Fig. 3c), and only 58-75% identity with each other, yet all bound to the stem helix mAbs by ELISA (Supplementary Fig. 14).

The binding affinities for the various ProteinMPNN designs based on the backbones of S2hlx-Ex2 (S2hlx-Ex15, -Ex17, -Ex19, and -Ex20), S2hlx-Ex3 (S2hlx-Ex46 and S2hlx-Ex54), S2hlx-Ex4 (S2hlx-Ex8-Trx and S2hlx-Ex25), and S2hlx-Ex6 (S2hlx-Ex34 and -Ex37) to CC40.8, S2P6, and DH1057.1 mAbs were determined by SPR (Supplementary Figs. 15, 16). S2hlx-Ex2 derived epitope scaffolds had the highest affinities with dissociation constants between 2 nM and 30 nM for S2P6, 2 nM and 49 nM for DH1057.1, and 3 nM to 27 nM for CC40.8 mAb (Fig. 3d; Supplementary Fig. 15). S2hlx-Ex15 and -Ex19 had affinities in a similar range to those measured for spike (Supplementary Fig. 8) and those reported for the stem helix peptide[9,10]. S2hlx-Ex54, derived from the backbone of S2hlx-Ex3, bound to S2P6, DH1057.1, and CC40.8 with equilibrium dissociation constants of 45 nM, 222 nM and 150 nM, respectively (Supplementary Fig. 16), while S2hlx-Ex37, based on the fold of S2hlx-Ex6, had affinities of 664 nM, 220 nM and 849 nM to the three target mAbs (Supplementary Fig. 16).

To test the ability of epitope scaffolds to engage diverse antibodies that may be elicited as part of a polyclonal response against the stem helix by vaccination, the designed antigens were tested against additional broadly reactive stem helix antibodies that were recently described and were not explicitly targeted at the design stage[13]. The epitope scaffolds bound all tested antibodies by ELISA (Fig. 3e; Supplementary Fig. 17a), indicative of their ability to recognize diverse antibodies that target the displayed epitope. In addition to the mature antibodies, binding was also tested to five of their UCA or inferred germline precursors (Fig. 3e; Supplementary Fig. 17b). Kinetic analysis revealed that S2hlx-Ex15 had low nanomolar affinities for DH1057 UCA and the iGLs for S2P6 and CC40.8 ($K_D$s of 22.4 nM 1.4 nM and 9.7 nM), and also bound strongly to Cov44-26 UCA and Cov89-22iGL by ELISA. Other S2hlx-Ex2 and S2hlx-Ex3 derived designs interacted similarly with three to five of the antibody precursors tested (Fig. 3d, Supplementary Fig. 15). Four out of the five antibody precursor recognized spike, but with significantly weaker affinity than S2hlx-Ex15 and S2hlx-Ex19 (Supplementary Fig. 17).

To ensure that the S2hlx epitope scaffolds interacted with the target mAbs in a similar fashion to spike, antibody binding was measured to eight epitope scaffolds that spanned all the design families (Fig. 3b) and that contained alanine mutations corresponding to spike epitope sites Leu1145 and Phe1148 in the stem helix (Supplementary Fig. 18). As anticipated based on structural analysis and previously published observations[10], all the epitope scaffolds lost binding to CC40.8 when the Leu1145 equivalent residue was mutated, while binding to S2P6 and DH1057.1 was not affected. When the epitope residue equivalent to spike Phe1148 was mutated, binding was lost to DH1057.1 and S2P6 in all designs, confirming that antibody interactions are mediated by the same residues on spike and the designed antigens.

Analysis of a high-resolution (2.3 Å) crystal structure of unbound S2hlx-Ex19 further confirmed the successful transplantation of the CC40.8 mAb epitope (Fig. 3f, Supplemental Table 1). As expected, the grafted epitope region in S2hlx-Ex19 diverged considerably from the backbone region it replaced in the parent scaffold (backbone RMSD = 5.8 Å), with the epitope adopting a more exposed conformation that allows for antibody engagement (Fig. 3f). This conformational change was likely facilitated by introducing two aromatic residues (Tyr62 and Phe66) under the epitope helix, which effectively pushed the epitope segment away from the rest of the protein core. Interestingly, the epitope conformation on S2hlx-Ex19 was almost identical to that present on the prefusion stem helix, rather than the more extended one induced by CC40.8 (Fig. 3f). Nevertheless, given the high

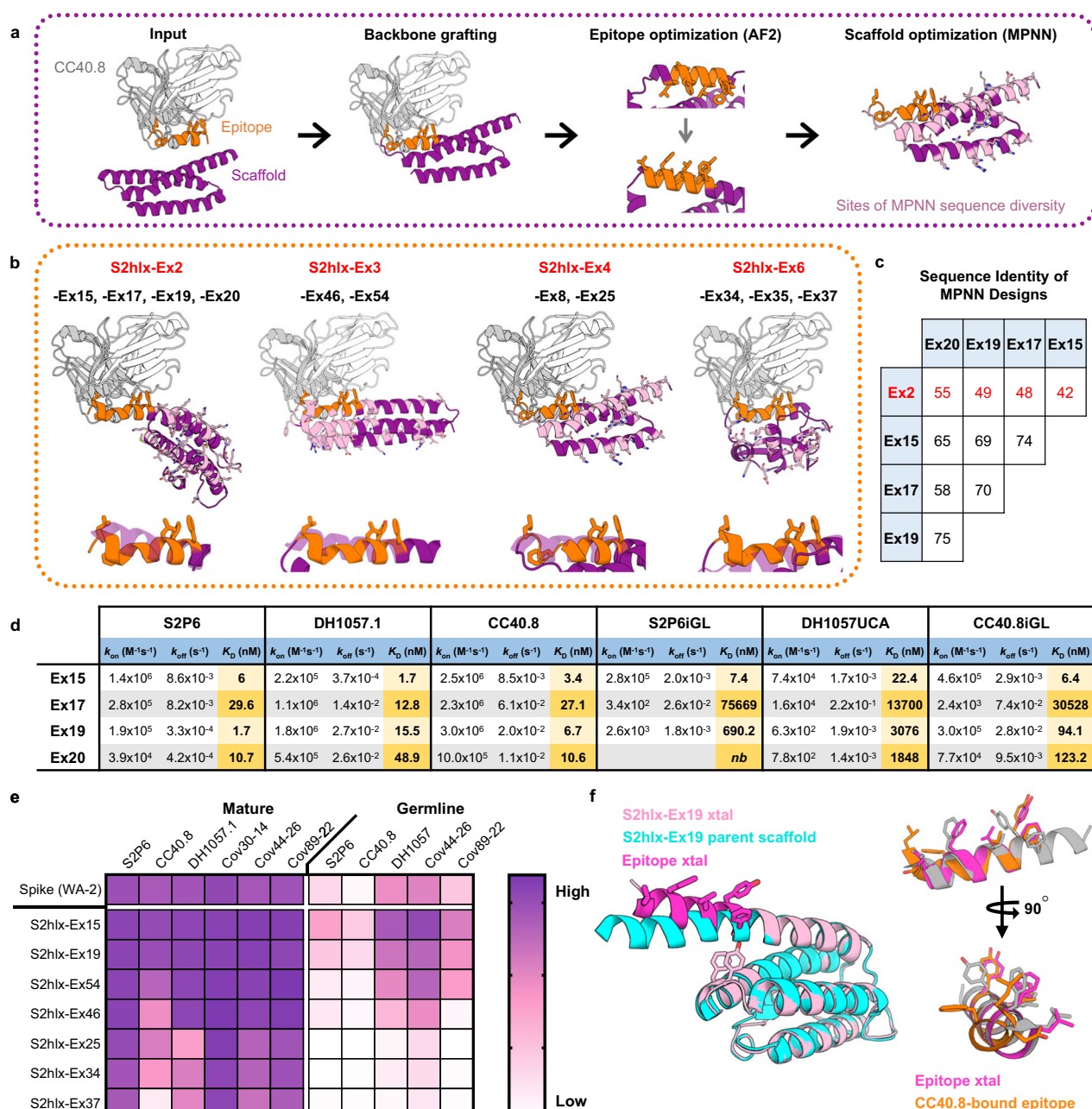

**Fig. 3 | Design of epitope scaffolds that engage two major classes of antibodies against the stem helix. a** Overview of the computational design process to graft the antibody-bound CC40.8 epitope onto candidate scaffolds. ESs were designed by transplanting the epitope backbone with Rosetta, validated with Alphafold2 (AF) and optimized for expression with ProteinMPNN (MPNN). **b** *Top*: Computational models of CC40.8 mAb (*gray and white*) bound to ESs (*purple*) from the S2hlx-Ex2, S2hlx-Ex4, and S2hlx-Ex6 families of designs. The epitope is highlighted in *orange sticks*. Residues designed by ProteinMPNN and that can differ between designs from the same family are shown in *pink sticks*. *Bottom*: Overlay of the grafted epitope conformation (*orange*) and the parent scaffold structure it replaced (*purple*). **c** Sequence identity between S2hlx-Ex2 family of ESs designed by ProteinMPNN. **d** Dissociation constants of S2hlx-Ex2 based designs to S2P6, DH1057 and CC40.8 mAbs and their precursors. *nb*=not binding. **e** Binding of spike and S2hlx-Ex ESs to diverse stem helix mAbs and their precursors. Source data are provided as a Source Data file. **f** *Left*: alignment of the S2hlx-Ex19 parent scaffold (*cyan*) with the crystal structure (*xtal*) of the unbound epitope scaffold (*pink*, with the epitope in *magenta sticks*). Residues Tyr62 and Phe66 that stabilize the grafted epitope conformation are highlighted. *Right*: Comparison of the stem helix epitope conformation in the pre-fusion spike (*gray*), induced by CC40.8 binding (*orange*) and displayed on S2hlx-Ex19 (*magenta*). Source data are provided as a Source Data file.

affinity of S2hlx-Ex19 for CC40.8 ($K_D = 6.7$ nM), it is likely that the antibody can readily induce the necessary conformation upon binding. Thus, by combining epitope backbone grafting and ProteinMPNN, we engineered antigens that bind with high affinity and specificity to the major classes of stem helix broadly reactive antibodies as well as to their inferred precursors.

## Cross-reactivity of engineered epitope scaffolds with pre-existing immune responses induced by SARS-CoV-2 spike

Next, we tested the ability of the engineered epitope scaffolds to cross-react with pre-existing immune responses elicited by SARS-CoV-2 spike in order to evaluate their potential to preferentially amplify spike[815–823] or stem helix directed responses by vaccination. Sera or

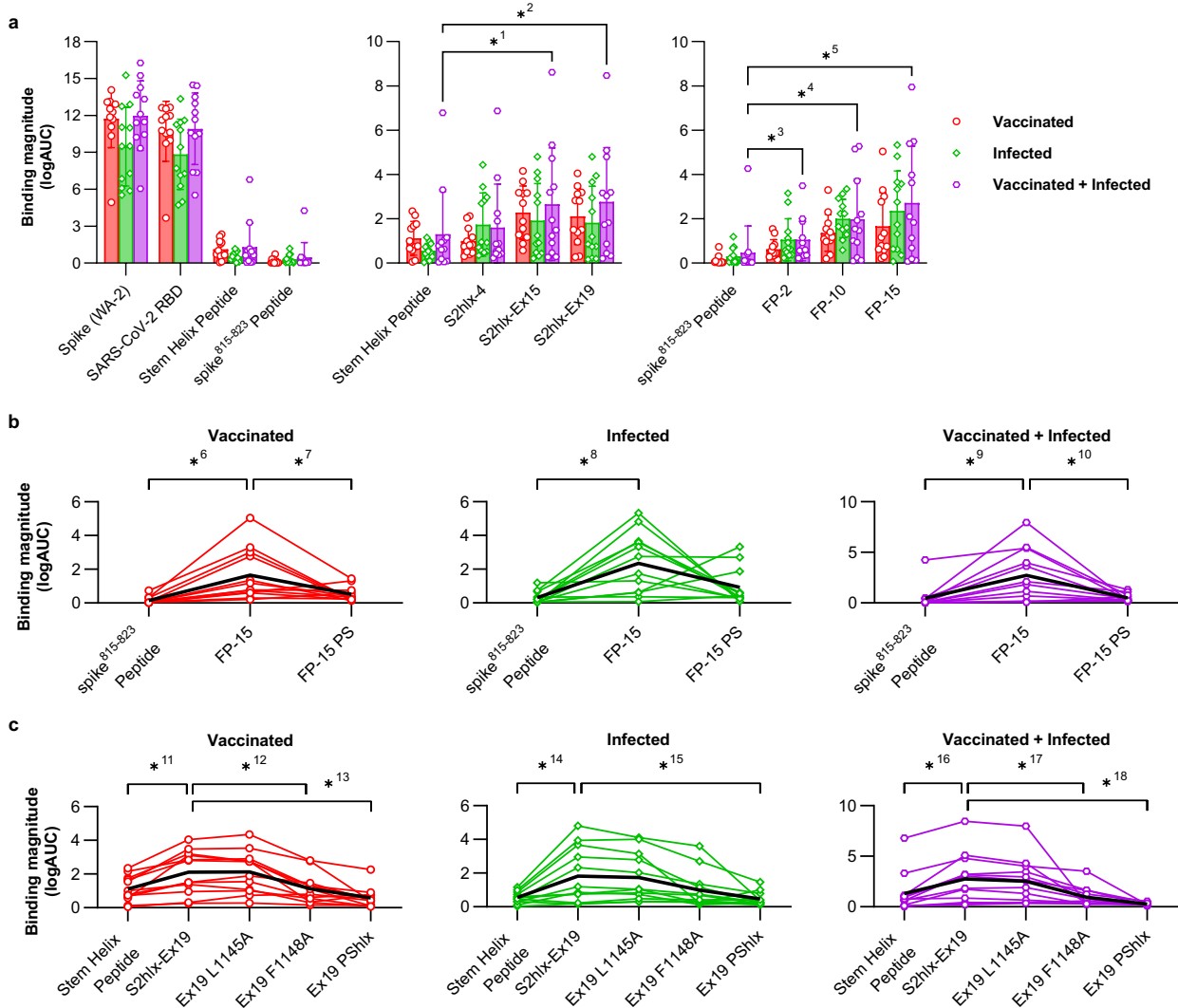

**Fig. 4 | Reactivity of engineered epitope scaffolds with sera from individuals with pre-existing immune responses to SARS-CoV-2 spike. a** *Left*: ELISA binding of sera or plasma from patients with pre-existing SARS-CoV-2 immunity acquired by vaccination (*red*), infection (*green*), or vaccination followed by infection (*purple*) to recombinant WA-2 spike and RBD or to synthetic peptides encoding the spike[815–823] peptide (residues 808-833) or the stem helix peptide (1140-1164). *Middle* and *right*: Binding of the same samples in (**a**) to select stem helix peptide and spike[815–823] peptide epitope scaffolds respectively. Data are presented as the mean +/- standard deviation. *[1]$p = 0.0005$, *[2]$p = 0.0005$, *[3]$p = 0.0269$, *[4]$p = 0.0005$, *[5]$p = 0.0005$. **b** Comparison between sera binding to the synthesized spike[815–823] peptide with FP-15 ES, and the parent scaffold from which FP-15 was derived and that lacks the grafted epitope (FP-15 PS). *[6]$p = 0.0005$, *[7]$p = 0.0161$, *[8]$p = 0.0005$, *[9]$p = 0.005$, *[10]$p = 0.0093$. **c** Comparison between sera binding to the synthesized stem helix peptide with S2hlx-Ex19 ES, and with versions of this design where the epitope is mutated to reduce binding to either CC40.8 class antibodies (L1145A), to both CC40.8 and S2P6 classes of antibodies (F1148A), or where the epitope is replaced with the sequence from the parent scaffold it replaced (PShlx). Single lines represent measurements for individual samples in the respective cohort. For all graphs $n = 12$ independent samples. Significance tested using the Wilcoxon signed-rank test. *[11]$p = 0.0005$, *[12]$p = 0.0034$, *[13]$p = 0.001$, *[14]$p = 0.0068$, *[15]$p = 0.0034$, *[16]$p = 0.005$, *[17]$p = 0.0034$, and *[18]$p = 0.0015$. Source data are provided as a Source Data file.

plasma samples isolated from three groups of 12 study participants who acquired SARS-CoV-2 immunity either through vaccination, infection, or vaccination followed by infection respectively, were first tested for binding to WA-2 spike, RBD and synthesized peptides encoding the spike[815–823] and stem helix regions (see Supplementary Table 2 for full patient information). This analysis revealed that only a small fraction of spike-directed humoral responses, regardless of their acquisition route, targeted the spike[815–823] or stem helix regions (Fig. 4a). On average, spike[815–823] and stem helix peptide binding levels were 23.3 and 8.4-fold lower respectively than RBD-focused responses across all groups, underscoring that the spike[815–823] and the stem helix domains are sub-dominant targets of antibodies elicited by spike.

Both spike[815–823] and S2hlx-Ex epitope scaffolds showed significantly higher recognition of human sera than the corresponding spike peptide domains (Fig. 4a, Supplementary Fig. 19). Among the spike[815–823] epitope scaffolds, FP-15 had the best sera recognition, showing 11.3, 7.7, and 5.8-fold higher IgG binding than the corresponding synthesized peptide in vaccinated, infected and hybrid immunity samples respectively (Fig. 4b). Importantly, these interactions were mediated by the engineered epitope contacts, as binding to the parent scaffold of FP-15 that does not contain the grafted epitope, was significantly reduced (3.5-fold in vaccinated, 2.5-fold in infected and 5.5-fold in hybrid, Fig. 4b). A similar trend was observed for FP-10 and FP-2, but their overall binding levels were lower than FP-15,

although still above those of the synthesized spike[815–823] peptide (Supplementary Fig. 19b).

From the stem helix designs, S2hlx-Ex19 and S2hlx-Ex15 had the best recognition of spike-induced immune responses (Supplementary Fig 19a). Compared to the synthesized stem helix peptide, S2hlx-Ex19 bound on average 2.4 times better across all three groups (Fig. 4c). The binding was epitope specific, as evidenced by the decrease in interactions with S2hlx-Ex19 variants that contained alanine mutations at hotspot epitope residues Leu1145 (LA) and Phe1148 (FA), or that had the whole epitope replaced by the sequence of the parent scaffold (S2hlx-Ex19 PShlx, Fig. 4c). As shown by the larger decrease in binding to the FA than LA mutants, cross-reactive spike responses were more dependent on stem helix residue Phe1148 which is recognized by both S2P6 and CC40.8 classes of antibodies (Fig. 4c, Supplementary Fig 19). Compared to S2hlx designs, the ability of the S2hlx-Ex epitope scaffolds to recognize CC40.8 mAb translated into a broader engagement of spike elicited humoral responses for these types of molecules (Fig. 4a, Supplementary Fig 19).

### Isolation of antibodies with broad coronavirus reactivity from humans pre-exposed to SARS-CoV-2 using epitope scaffolds

Next, we characterized the ability of the engineered epitope scaffolds to interact with B cells that give rise to antibodies with broad activity against the spike[815–823] peptide or the stem helix from people with preexisting SARS-CoV-2 immunity, as a way to evaluate the potential of our immunogens to boost such responses by vaccination. Stem helix reactive BCRs were isolated from IgD- B cells, a pool that included both memory B cells and plasmablasts, from the two subjects with the highest sera titers against S2hlx-Ex19, using positive selection for S2hlx-Ex19 and negative selection for the parent scaffold S2hlx-Ex19-PS that lacks the epitope (Fig. 5a, b). Stem helix epitope-specific memory B cells were isolated at a frequency of ~1:3000 and ~1:2100 respectively from the two samples (Fig. 5b, Supplementary Fig 20).

Antibody sequences were amplified from the isolated B cells and preliminarily screened for spike binding[22] in order to select monoclonal antibodies for recombinant production and in-depth functional characterization. Twelve recombinant antibodies selected from B cells that bound S2hlx-Ex19 but not S2hlx-Ex19-PS were tested for binding against a panel of spike proteins from SARS-CoV-2 variants as well as other human and animal betacoronaviruses. Ten antibodies showed measurable affinity to multiple coronavirus spikes, with three of them displaying binding on par to that of S2P6 and CC40.8 mAbs to all the spikes tested, including SARS-CoV-1, MERS, OC43 and HKU-1 (Fig. 5c, Supplementary Figs. 21–24). Interestingly, these three antibodies, DH1501.1, DH1501.2, and DH1501.3, were clonally related and their heavy and light chain V gene segment pairing (IGHV1-46/IGKV3-20) matched those of S2P6 mAb. The inferred UCA of these antibodies displayed tight binding to multiple spikes, similar to what was reported for the iGL of S2P6 mAb (Supplementary Fig. 25). Beyond the IGHV1-46/IGKV3-20 derived mAbs, multiple other unique $V_H/V_L$ pairings, not previously reported, were observed in the isolated antibodies (Fig. 5c). This suggests the presence of a clonally diverse response against the stem helix epitope that can be engaged by the engineered epitope scaffold. The best six antibodies by affinity and breadth were tested side by side with S2P6 and CC40.8 for their ability to neutralize multiple SARS-CoV-2 variants and SARS-CoV-1 in a pseudovirus neutralization assay. CC40.8 and two isolated mAbs, DH1501.1 and DH1501.3, were the only ones that displayed neutralization activity, albeit with high $IC_{50}$ values above 1 μg/mL that are typical for antibodies against this epitope. DH1501.3 and CC40.8 neutralized all the pseudoviruses in the panel, including XBB1.5 and SARS-CoV-1 (Fig. 5d).

Given that stem helix antibodies were previously shown to protect against infection in animal models despite limited neutralization potency[9,10], we next investigated whether the isolated monoclonal antibodies could control the virus through antibody-mediated

functions. To determine the ability of these mAbs to mediate Fc effector functions, they were first tested for binding to virus-infected cells, a prerequisite of antibody-dependent cellular cytotoxicity (ADCC). Three of the five antibodies tested, DH1501.1, DH1501.2, and DH1501.3, showed binding to cells infected with either the D614G or the BA.1 variant of SARS-CoV-2 in the same range as the previously described stem helix antibodies CC40.8, S2P6 and DH1057.1 (Fig. 5e). Binding of the stem helix antibodies was lower that of the RBD-directed mAb DH1047, supporting the observation that the stem helix epitope is more occluded on the viral spike. Next, we performed a Natural Killer (NK) cell degranulation assay as a surrogate for assessing ADCC activity. Antibodies DH1489, DH1501.1, DH1501.2, and DH1501.3 induced NK cell degranulation against all three variants of concern tested, with potencies similar to those of known stem helix antibodies (Fig. 5f). Interestingly, all stem helix mAbs induced the most potent degranulation against BA.4/5 spike-transfected cells. Overall, isolated antibodies with IGHV1-46/IGKV3-20 immunogenetics had the highest activity in both the pseudovirus neutralization and ADCC assays. While other isolated antibodies with different $V_H/V_L$ pairings were not as broad or potent, it is possible that they could be further matured to breadth by targeted vaccination and may be valuable targets for rational vaccine design.

We similarly isolated antibodies against the spike[815–823] epitope by sorting B cells that bound the FP-10 epitope scaffolds and lacked binding to the original parent scaffold that did not contain the grafted epitope. The frequency of epitope specific B cells selected was ~1:4,000 in two analyzed samples and two antibody sequences were recovered for recombinant expression and characterizations after preliminary screening. These antibodies had distinct immunogenetics not observed in other previously isolated antibodies against the spike[815–823] epitope (Fig. 5c). Nevertheless, they bound tightly to diverse coronavirus spikes, including those of human alpha coronaviruses 229E and NL63 as is typical of antibodies with broad activity against this epitope (Fig. 5c). No binding was detected for SARS-CoV-2 variants BA.1 and XBB1.5, however these recombinant spikes contained a "hexapro" mutation that is known to affect the presentation of the spike[815–823] epitope and to limit the binding of antibodies against this site[8].

Antibody isolation from subjects with pre-existing SARS-CoV-2 immunity revealed that the engineered epitope scaffolds interact with diverse monoclonals against the target epitopes. Some of the antibodies are derived from the same $V_H$ and $V_L$ genes observed in antibodies previously isolated[9]. While the epitope scaffold reactive mAbs against the stem helix had weak neutralization potency, they exhibited strong ADCC activity in vitro, revealing a potential protection mechanism consistent with previous observations for this antibody class. Taken together, the robust sera binding and memory B cell engagement by epitope scaffolds demonstrate their strong recognition of pre-existing SARS-CoV-2 immune response and support their use for boosting spike[815–823] and stem helix antibodies by vaccination.

### Engineered stem helix epitope scaffolds elicit antibodies with broad reactivity against diverse CoVs by vaccination

Next, we assessed the ability of the engineered epitope scaffolds to elicit antibodies against the target epitope by vaccination. We focused on testing the stem helix immunogens belonging to the S2hlx-Ex2 family of design, Ex15, Ex19, Ex20, and Ex17, because: 1) they have high affinities for all the antibodies tested as well as their precursors; and 2) while they share the same backbone structure, their sequence outside the epitope is highly divergent, which may help focus immune responses on the epitope by vaccinating with combinations of different designs.

Epitope scaffolds were multimerized on mi03 nanoparticles (NPs) using the SpyCatcher/Spytag[23] conjugation system in order to improve immune presentation (Fig. 6a). Homotypic nanoparticles were

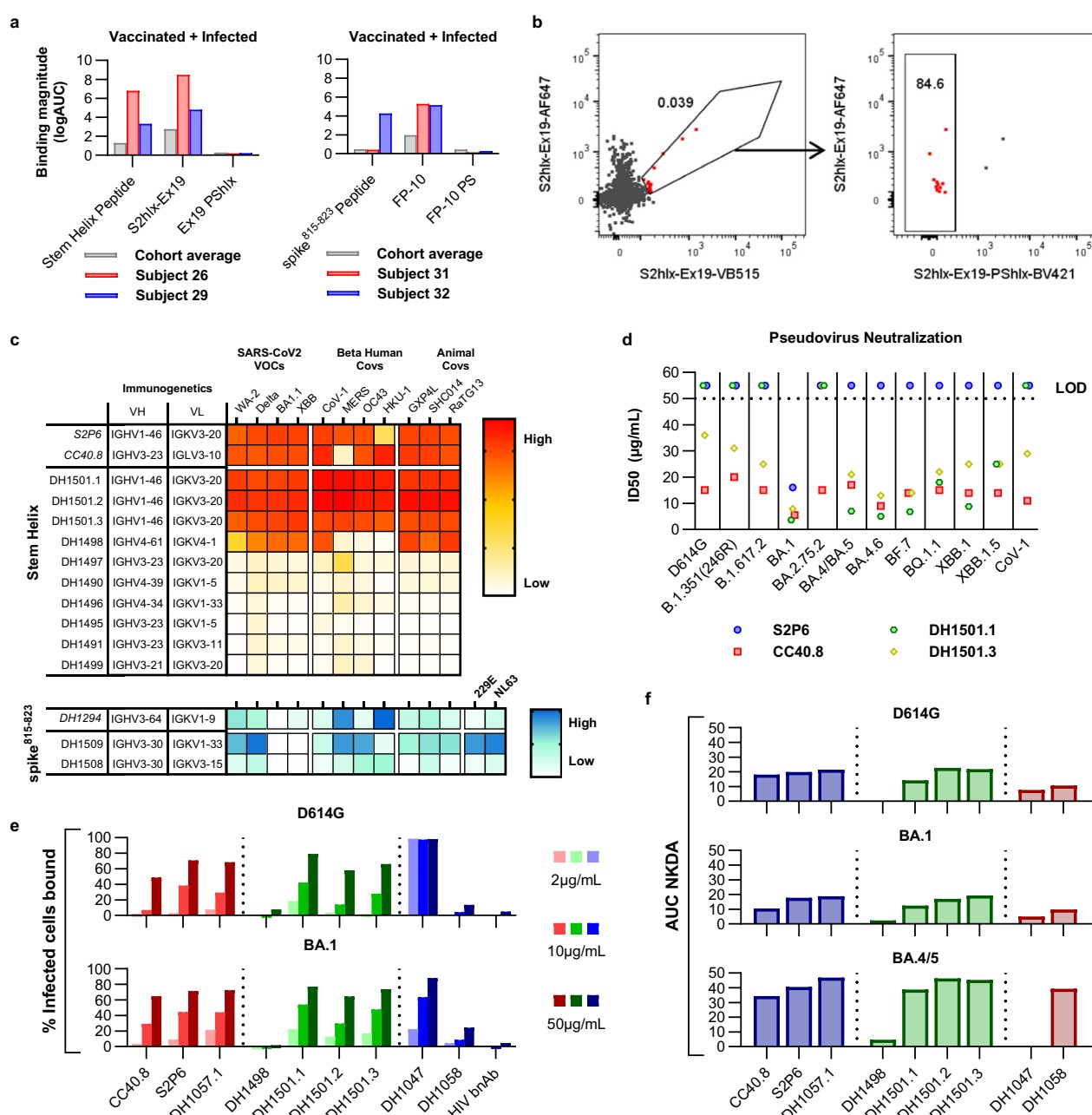

**Fig. 5 | Isolation and characterization of epitope scaffold reactive antibodies from subjects with pre-existing SARS-CoV-2 immunity. a** Sera binding profile of subjects from which B cells were isolated. Binding was measured to the synthetic peptide epitope, the epitope scaffold used as the sorting probe, and the corresponding parent scaffold (PS) that lacks the epitope and was employed for negative selection. **b** Representative FACS plots of memory B cells labeled with epitope scaffold probes. Cells from Subject 26 were labeled with S2hlx-Ex19 conjugated to VB15 (x-axis) and AF647 (y-axis). A third bait comprising S2hlx-Ex19 PShlx conjugated to BV421 was used to exclude scaffold-specific memory B cells. **c** Binding of the isolated antibodies to diverse CoV spikes. **d** Neutralization of diverse pseudoviruses by isolated stem helix antibodies. CoV-1 = SARS-CoV-1. All other pseudoviruses are the indicated variant of SARS-CoV-2. **e** Binding of isolated antibodies to cells infected with two different SARS-CoV-2 VOCs (D614G, *top*; BA.1, *bottom*). Antibodies in *red* have been previously described, those depicted in *green* were isolated here, while those in *blue* represent controls; DH1047 targets the RBD domain; DH1058 targets the spike[815–823] peptide domain; anti-HIV antibody VRC01 was included as a negative control. **f** NK cell degranulation ability of mAbs from (**e**). Source data are provided as a Source Data file.

developed that displayed 60 copies of designs Ex15, Ex19, or Ex20 respectively. A "mosaic" NP was also engineered where Ex15, Ex19, Ex20, and Ex17 were conjugated together. Antigenic and NSEM characterization confirmed that the NPs were well formed and bound S2P6, DH1057.1 and CC40.8 (Supplementary Fig. 26). Groups of eight BALB/c mice were vaccinated three times, four weeks apart with either: 1) Ex15-NP; 2) Ex_mosaic-NP; 3) a mixture of individual Ex15-NP, Ex19-NP and Ex20-NP; 4) a sequential regimen of Ex15-NP prime, followed by Ex19-

NP and Ex20-NP boosts; or 5) GLA-SE only, the adjuvant included in all the immunizations, as control (Fig. 6b). After two immunizations, sera from all animals vaccinated with epitope scaffolds showed strong reactivity to WA-2 and XBB SARS-CoV-2 spikes (Fig. 6c). Sera breadth was tested against spikes from all human beta coronaviruses as well as RsSHC014 and GXP4L, two pre-emergent animal viruses from bats and pangolins. Binding was observed against all the spikes tested, with the highest activity against SARS-CoV-1, RsSHC014, and GXP4L (Fig. 6d).

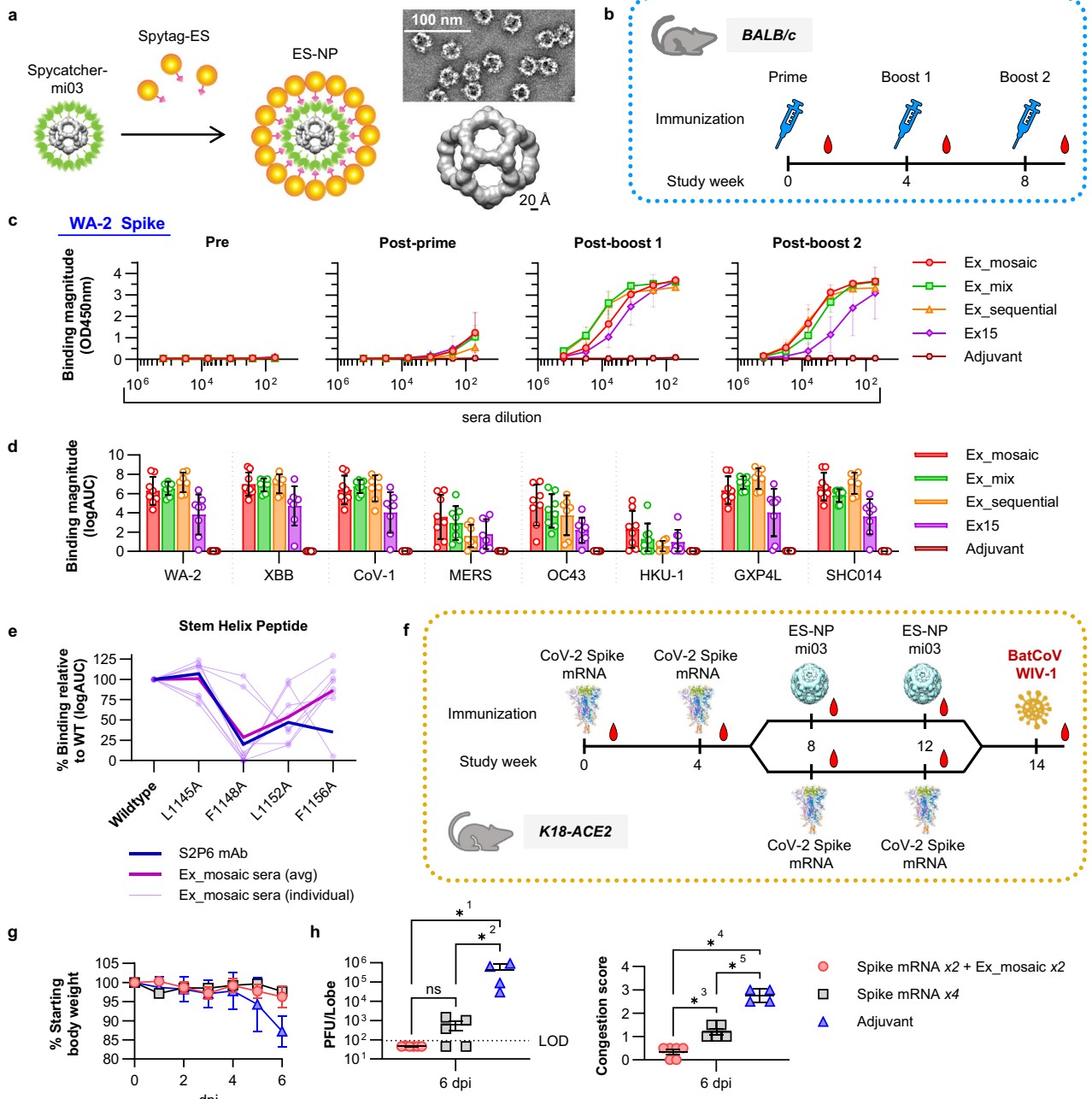

**Fig. 6 | Immunogenicity of stem helix epitope scaffolds. a** Development and NSEM analysis of the spycatcher/spytag mi03 nanoparticles (NP) displaying 60 epitope scaffold copies. **b** Study design to determine the immunogenicity of stem helix nanoparticles in BALB/c mice. **c** Binding of sera from animals immunized with the different epitope scaffold regimens against the SARS-CoV-2 WA-2 spike at different study time points. WA-2 and XBB refer to the SARS-CoV-2 variants. CoV-1 = SARS-CoV-1. GXP4L= Pangolin CoV GX-P4L. SHC014= Bat CoV RsSHC014. **d** Sera binding after the second boost against diverse CoV spikes. Values are calculated as the logarithm of the area under the curve for binding curves measured as in (**c**). **e** Binding of S2P6 mAb and sera from animals immunized with Ex_mosaic to the WT stem helix epitope peptide and mutated variants. **f** Study design to determine the protection ability of Ex_mosaic-NP against live coronavirus challenge. **g** Percentage weight loss in animals treated as in (**f**). Data are presented as the mean +/- standard deviation. **h** Lung congestion score and plaque analysis of infected animals. For **g** and **h**, adjuvant *n* = 4, Spike mRNA x4 *n* = 5, Spike mRNA x2 + Ex_mosaic x2 *n* = 6. Data points correspond to individual animals. F was calculated by one way ANOVA using Brown-Forsythe test and comparisons using Tukeys test. ns *p* = 0.9999; *[1]*p* = 0.027; *[2]*p* = 0.033; *[3]*p* = 0.005 **[4]*p* = 0.00001; *[5]*p* = 0.00005. Source data are provided as a Source Data file.

Importantly, sera cross-reacted with OC43, MERS, and HKU-1 spikes, although at lower levels than those measured against SARS-CoV-2, but consistent with the induction of a broad response against the stem helix epitope (Fig. 6d). Of the four immunization regimens tested, the homotypic Ex15 NP elicited the lowest titers and breadth. Combinations of epitope scaffolds whether administered together, in mosaic form or sequentially, performed similarly well, although the

Ex_mosaic-NP elicited the highest average titers against heterologous human betacoronaviruses.

Epitope specificity was demonstrated by strong sera binding to an unrelated epitope scaffold not used in the immunizations (Supplementary Fig. 27). Epitope specificity was further demonstrated by measuring binding to synthesized peptides encoding the stem helix region as well as mutated variants at sites known to be important for

antibody recognition (Fig. 6e). Mutations at residues 1148, 1152, and 1156 decreased sera binding, while alanine substitution at position 1145 had limited effect, consistent with the elicitation of antibodies that resemble the binding mode of S2P6 or DH1057.1, rather than that of CC40.8 (Fig. 6e). No neutralizing activity was observed for the elicited sera, which was unsurprising given the limited activity of S2P6-like recombinant monoclonals in our pseudovirus neutralization assay (Fig. 5d).

Next, we assessed the ability of the Ex_mosaic-NP to boost stem helix specific responses in animals previously vaccinated with mRNA encoding the SARS-CoV-2 WA-2 spike and to protect against a viral challenge with a heterologous virus. Assessing protection was particularly important since sera elicited by stem helix epitope scaffolds did not neutralize in our assay; however, antibodies that react with the grafted epitope can promote strong ADCC as shown above (Fig. 5e, f) and may protect through antibody-mediated functions. K18-ACE2 mice were immunized every four weeks with either four shots of mRNA spike or twice with mRNA spike followed by two boosts of the Ex_mosaic-NP. Two weeks after the last immunization, the animals were challenged with live WIV-1 virus (Fig. 6f). WIV-1 was chosen because its spike sequence is significantly different from that of SARS-CoV-2 (78% over the whole spike and 75% over RBD), while the stem helix epitope sequence is conserved, as in most sarbecoviruses. Animals vaccinated with Ex_mosaic-NP had higher average titers of antibodies targeting the stem helix epitope compared to those that received only spike mRNA, as measured by binding to an epitope scaffold not used in the immunization; binding to the synthesized stem helix peptide was also higher, but not statistically significant (Supplementary Fig. 28). Both immunizations regimens provided strong protection from WIV-1 based on the percentage of body weight the animals lost six days after the challenge and when compared to animals that received adjuvant only (Fig. 6g). However, animals boosted with Ex_mosaic-NP scored better on two other clinical measures compared to those that received spike mRNA only. No viral RNA was detected in the lungs of any of the Ex_mosaic-NP immunized animals and their congestion scores were significantly lower when compared to those of mRNA spike vaccinated animals (Fig. 6h). These results demonstrate that stem helix epitopes elicit sera with broad reactivity against beta coronaviruses and can offer protection against divergent viruses when used to boost immune responses initially induced by mRNA spike immunizations.

## Discussion

While currently approved coronavirus vaccines and multiple vaccine candidates are based on spike or RBD proteins[24], here we describe new types of immunogens that aim to elicit different humoral responses, focused on the conserved stem helix or the spike[815–823] epitopes in the S2 domain of spike. The ability to elicit high titers of antibodies against these regions will likely be critical to the development of a pan-coronavirus vaccine[25], a goal the molecules designed here aim to contribute towards. The epitope scaffold proteins were engineered using a combination of previously developed grafting methods[15,26] and more recently described machine learning techniques[21] that proved critical for improving both immunogen stability and antigenicity. Structural and mutational analysis confirmed that the epitope scaffolds engage multiple antibodies, as well as their precursors, with high affinity and specificity. For each class of epitope scaffolds, some designs showed tighter antibody binding than corresponding epitope peptides, especially as it relates to the recognition of antibody precursors.

Spike[815–823] and stem helix epitope scaffolds that bound S2P6 and DH1058 mAbs, but not CC40.8 mAb, were engineered using side chain grafting, an approach that modifies the sequence of existing proteins to accommodate the epitope and that was successfully used in the past to engineer epitope scaffolds against HIV[16,17,27] and RSV[28] antibodies.

However, for conformationally complex epitopes like the one induced by the CC40.8 mAb on spike, no known protein structures may exist that are structurally similar to allow side chain grafting. To successfully transplant these types of epitopes, different methods, called flexible backbone approaches, are needed to model and control the conformation of the epitope on the target scaffolds. This can be achieved either by engineering epitope scaffolds de novo by folding idealized protein backbones around the target epitope, as was previously done for epitopes on RSV[18,29], or by modifying the backbone of existing proteins to adopt the conformation of the grafted epitope, as illustrated before for HIV[26,27] and here for designs that bind CC40.8 mAb. Previously, the success of flexible backbone epitope grafting was limited; the majority of the computational designs failed to express recombinantly and the ones that did typically required multiple rounds of directed evolution to achieve high affinities to the target antibodies.

Here we combined previously described backbone grafting with recent machine learning based protein engineering approaches to significantly improve the success rate of epitope scaffolds designed using flexible backbone modeling. Structure prediction with AlphaFold2[20] was used to validate the conformation of the grafted epitope in the structural context of the target scaffold, while ProteinMPNN[21] was subsequently employed to identify optimal sequences that fold into the desired epitope scaffold structure. With this approach, multiple proteins were engineered in silico that bound tightly to CC40.8 mAb without needing additional experimental optimization. Of note, the epitope was successfully transplanted on scaffolds with diverse backbone structures, supporting our approach as a general way to present structurally complex motifs in diverse molecular contexts. Development of stem helix epitope scaffolds that bound with high affinity to all three target antibodies (S2P6, DH1057.1, CC40.8) proved challenging because CC40.8 induces a different conformation on the epitope upon binding compared to the other two antibodies. This suggests that, for a successful design, the epitope presentation must allow for antibody-induced fit and some level of conformational changes. This was not explicitly modeled here at the design stage but likely represents an area of improvement in the computational workflow for transplanting epitopes that adopt different antibody-bound conformations in the future. Interestingly, the stem helix immunogen S2hlx-Ex19, which bound with affinities for CC40.8 and S2P6 similar to those of the native spike, displays the epitope at the N-terminus of the scaffold. While a crystal structure of this unbound epitope scaffold revealed that the epitope is well folded and adopts a helical conformation, it is possible that its positioning may more readily allow for conformational changes upon antibody binding relative to other designs where the epitope is grafted internally.

The ability of engineered immunogens to engage diverse antibodies against the displayed epitope is likely important for their ability to elicit a robust polyclonal response in vivo, as has been recently demonstrated for HIV vaccine candidates[30,31]. The epitope scaffolds designed here engaged multiple spike[815–823] and stem helix antibodies with high affinity in vitro, which translated to robust reactivity to polyclonal sera elicited by SARS-CoV-2 and the engagement of BCRs with diverse immunogenetics against the target epitopes. The stem helix epitope scaffolds that also bound CC40.8 mAb in addition to S2P6 and DH1057.1 mAbs, showed the highest level of binding to human sera from subjects exposed to SARS-CoV-2 by vaccination, infection or both. The S2hlx-Ex19 design was used to isolate multiple antibodies with broad reactivity against the stem helix from people with pre-existing immunity. Some of these monoclonal antibodies had the same $V_H/V_L$ pairings (IGHV1-46/IGKV3-20) as previously isolated antibodies, suggesting that this type of antibody is commonly induced by SARS-CoV-2 immunization and may be targeted for boosting with a next-generation vaccine. However, as found here, other broad

antibodies with different immunogenetic characteristics exist against S2 and they can be engaged by the designed epitope scaffolds as well. Isolated stem helix antibodies displayed limited neutralization against SARS-CoV-2 variants, consistent with previous reports describing other antibodies against this epitope[9,14]. Interestingly, despite their limited neutralization, stem helix antibodies were shown by others to protect well against live virus challenges[9,14], likely through their antibody-mediated functions which we demonstrated our antibodies also have.

In initial immunization studies, stem helix epitope scaffolds elicited sera with broad reactivity against all human beta coronaviruses and towards two animal viruses tested. Sera bound specifically to the target epitope in a manner consistent with the induction of S2P6-like mAbs. Future studies will analyze the B cell repertoire of vaccinated animals in more detail to better understand the specificity and protection ability of the elicited responses. The induced sera did not neutralize SARS-CoV-2 pseudoviruses in our assay, consistent with our observations and previous reports that antibodies against the stem helix are poor neutralizers but can protect through antibody mediated functions[9,14]. To demonstrate that the engineered immunogens can protect against coronavirus infection, we vaccinated K18-ACE mice pre-exposed to spike mRNA with stem helix immunogens, prior to a viral challenge with WIV-1. Compared to an adjuvant-only group, mice boosted with our immunogens were protected against weight loss, had limited lung congestion, and undetectable viral load in the lungs. While mice immunized with only mRNA spike were also protected against weight loss, they exhibited increased congestion and higher titers of lungs viremia compared to mice that received the stem helix epitope scaffolds. Taken together, these results support the continued development and characterization of epitope scaffold immunogens to boost responses against conserved S2 epitopes as part of a next-generation pan betacoronavirus vaccine.

## Methods

### Computational design of epitope scaffolds

**Design of epitope scaffolds by side chain grafting.** A database of 9884 candidate scaffolds was created by selecting from the Protein Data Bank (PDB) structures that were: 1) determined by x-ray crystallography; 2) high resolution (<2.8 Å); 3) monomeric; 4) expressed in *E. coli*; 5) not of human origin and 6) that did not contain ligands. Epitope scaffolds were designed with Rosetta as previously reported, using the RosettaScripts for side chain grafting described in *Silva et. al.*, with the following parameters in the *MotifGraft* mover: RMSD_tolerance = "0.3"; NC_points_RMSD_tolerance = "0.5"; clash_test_residue = "ALA"; clash_score_cutoff = "5". For the design of spike[815–823] peptide (FP) epitope scaffolds, the structure of spike[815–823] peptide fragment [815]RSFIEDLLF[823] as described in the crystal structure of the DH1058-FP peptide complex (PDBid: 7tow) was used as the target epitope to graft. With the exception of residues F817 and L821 which were allowed to change in the designed epitope scaffold because they did not contribute to antibody binding, the identity of all the other epitope residues was maintained. For the design of epitope scaffolds that bind S2P6 and DH1057.1 mAbs, the structure of the stem helix fragment [1148]FKEELDKYF[1156 from] PDBid:7rnj was grafted. The identity of residues E1150 and K1154 was allowed to change during the design process while the other epitope residues were kept fixed. From the epitope scaffold generated by the automated protocol, the top 100 models by Rosetta ddG that were also smaller than 150 amino acids were visually examined to ensure appropriate epitope transplantation and antibody-scaffold interaction. At this stage, the suitability of a given parent scaffold in terms of function, conformational flexibility, and expression protocol was also investigated by referencing the publication it originated from. If necessary, additional changes were introduced in a candidate epitope scaffold using Rosetta fixed backbone design to remove antibody-epitope scaffold contacts that

were not due to the target epitope and to ensure proper interactions between the epitope and the rest of the scaffold. The best 15 designs for each of the target epitopes were chosen for experimental characterization.

### Design of stem helix epitope scaffolds by backbone grafting and MPNN optimization

Epitope scaffolds that display the stem helix epitope recognized by the CC40.8 antibody (PDBid: 7sjs) were designed by backbone grafting as previously reported and using the RosettaScripts described by Silva et. al. The same set of curated parent scaffolds as the one used for side chain grafting above was searched here. The structure of the stem helix fragment [1144]ELDSFKEELDKYFK[1157] from PDBid:7sjs was grafted. The identity of residues E1144, D1146, S1147, E1150, K1154, and K1157 were allowed to change during the design process while the other epitope residues were kept fixed. The following parameters were used:

RMSD_tolerance = "5.0" NC_points_RMSD_tolerance = "0.75"
clash_score_cutoff = "5" clash_test_residue = "ALA"
hotspots = "2:5:6:8:9:10:12:13"
combinatory_fragment_size_delta = "1:1"
max_fragment_replacement_size_delta = "−4:4"
full_motif_bb_alignment = "0"
allow_independent_alignment_per_fragment = "0"
graft_only_hotspots_by_replacement = "0"
only_allow_if_N_point_match_aa_identity = "0"
only_allow_if_C_point_match_aa_identity = "0"
revert_graft_to_native_sequence = "0"
allow_repeat_same_graft_output = "0" />

The initial hits were filtered to remove scaffolds with (1) an average clash score greater than 5 across the epitope, and (2) a Rosetta calculated ddg greater than 0. The top 100 models by Rosetta ddG that were also smaller than 150 amino acids were then visually examined as above and then Alphafold2 was used to predict the structure of the 10 best designs using the following parameters-

--model_preset=monomer
--db_preset=full_dbs
--max_template_date=2021-11-01
--uniref90_database_path = /data/uniref90/uniref90.fasta
--mgnify_database_path = /data/mgnify/mgy_clusters.fa
--uniclust30_database_path = /data/uniclust30/uni-
  clust30_2018_08/uniclust30_2018_08
--bfd_database_path = /data/bfd/
  bfd_metaclust_clu_complete_id30_c90_final_seq.sorted_opt
--template_mmcif_dir = /data/pdb_mmcif/mmcif_files
--obsolete_pdbs_path = /data/pdb_mmcif/obsolete.dat
--pdb70_database_path = /data/pdb70/pdb70

The scaffolds were then aligned via the grafted epitope to the stem helix fragment in the CC40.8 structure PDBid:7sjs and inspected for clashes between the scaffold and antibody. Scaffolds with clashes, such as where the antibody binding epitope was occluded or buried, were modified such that key epitope residues above remained fixed while other residues in the epitope and in contacting parts of the scaffold were allowed to vary through iterative rounds of fixed-backbone rotamer-based sequence design in Rosetta. After each round, we checked the Alphafold2 prediction and repeated until the epitope was predicted to be presented in a suitable orientation for antibody binding (Fig. 3a). Expression tags were screened by N-terminally tagging S2hlx-Ex4 with GST, SUMO, Thioredoxin (Trx), Maltose binding protein (MBP), or F8H tags. As Trx gave the highest fraction of full-length protein it was then used to tag S2hlx-Ex2 and -Ex6 (Supplementary Fig. 11).

The parent structure of S2hlx-Ex4 (PDBid:2QYW) was used to screen the VAST+ database (https://structure.ncbi.nlm.nih.gov/Structure/VAST/vast.shtml) for homologous structures, which were identified by RMSD (<2 Å) and high fraction of alignment (>80%). Two

homologs identified were expressed with Trx tags (Supplementary Fig. 12).

For ESs where protein expression was low the protein structures, either the Rosetta model or Alphafold prediction, were entered into ProteinMPNN at the website, https://huggingface.co/spaces/simonduerr/ProteinMPNN. Initial designs for S2hlx-Ex2 and -4 used sampling temperature of 0.1 and backbone noise of 0.02 and the whole epitope sequence was fixed, For S2hlx-Ex2, this approach produced six related soluble ESs but failed for S2hlx-Ex4. The second set of S2hlx-Ex4 and S2hlx-Ex6 designs used the higher sampling temperature (0.25) and higher backbone noise (0.2) to increase sequence diversity in designs, and only $L^{1145}xxFKxELDxYF^{1156}$ residues were fixed. ES structures were modeled using the integrated Colabfold prediction and scored by RMSD and pLDDT, and the highest scoring were re-predicted using Alphafold2. For the S2hlx-Ex6 designs, the epitope was occluded or buried in several designs. In these cases, residues in the epitope or surrounding parts of the ES were manually modified and the ES structure prediction repeated using Alphafold2 until the epitope was predicted to be suitable to binding to S2P6. Figures showing structures were prepared in Pymol v2.5.5 (Schroedinger).

### Plasmids and DNA synthesis

Genes encoding designed epitope scaffolds were commercially synthesized and cloned into pET29b (Genscript). Genes encoding the antibody heavy and light chains were similarly synthesized and cloned into the pcDNA3.1 vector (GenScript). Oligonucleotides were synthesized by IDT. Mutations were introduced in the synthesized plasmids by Q5 mutagenesis (NEB).

### Recombinant protein expression and purification

**Epitope scaffolds.** Plasmids encoding epitope scaffolds were transformed into *E. coli* BL-21 (DE3) (New England Biolabs) cells and 5 mL starter cultures were grown overnight at 37 C in Lysogeny Broth (LB) supplemented with 50 μg/mL kanamycin. The cultures were diluted 1:100 in Terrific BrothTB media (RPI) supplemented with kanamycin and grown at 37 °C to an OD600 of ~0.6. The temperature was subsequently lowered to 16 °C and the cultures were grown to OD600 of ~0.8 and induced with isopropyl β-D-1-thiogalactopyranoside (IPTG) to a final concentration of 0.5 mM. The cultures were shaken for 16–18 hours at 200 rpm. Cell pellets were collected by centrifugation at $14,500 \times g$ and lysed in B-PER Reagent (ThermoFisher Scientific) according to the manufacturer's protocol. The lysates were centrifuged at $14,000 \times g$ for 30 minutes at 4 C. The supernatant was incubated with Ni-NTA beads (Qiagen) equilibrated with native binding buffer (20 mM NaH2PO4, 500 M NaCl, 2% glycerol, 10 mM imidazole, pH=7.5) for an hour at 4 C. The beads were settled by centrifugation and the supernatant was removed by pipetting. The beads were washed with wash buffer (50 mM NaH2PO4, 500 M NaCl, 2% glycerol, 30 mM imidazole, pH=7.5) after which the protein was eluted with elution buffer (20 mM NaH2PO4, 500 M NaCl, 250 mM imidazole, pH=7.5). Protein expression and purity was confirmed by SDS-PAGE analysis, and quantified spectrophotochemically at 280 nm on a Nanodrop 2000 (ThermoFisher Scientific). The eluted protein was concentrated on 3 kDa MW spinMW spin columns (ThermoFisher Scientific) and further purified by size exclusion chromatography in 20 mM NaH2PO4, 150 mM NaCl, pH=7.5, on an AKTA-Go FPLC (Cytiva) using a Superdex 200 Increase 10/300GL column. Fractions containing monomeric protein were pooled and concentrated as above.

**Monoclonal antibodies.** Expi293F cells (ThermoFisher Scientific) were split to a density of $2.5 \times 10^6$ cells/mL in Expi293 Expression Medium with GlutaMAX (Gibco) and were transiently transfected with an equimolar plasmid mixture of heavy and light chain using Expifectamine (Invitrogen). For a typical 100 mL size transfection 100 μg amount of total DNA and 270 μL of lipofectamine were used. After overnight incubation, enhancers were added as per the manufacturer's protocol and the cultures were incubated with shaking for five days at 37 C and 5% CO₂. The cell culture was centrifuged to remove the cells and the supernatant was filtered with a 0.8-micron filter. The filtered supernatant was incubated with equilibrated Protein A beads (ThermoFisher) for one hour at 4 °C and washed with 20 mM Tris, 350 mM NaCl at pH=7. The antibodies were eluted with a 2.5% Glacial Acetic Acid Elution Buffer and were buffer exchanged into 25 mM Citric Acid, 125 mM NaCl buffer at pH=6. IgG expression was confirmed by reducing SDS-PAGE and quantified by measuring absorbance at 280nmm (Nanodrop 2000).

### Antigenic characterization of epitope-scaffolds

**Binding by surface plasmon resonance.** To determine the dissociation constants ($K_D$s) between epitope scaffolds and target antibodies, binding was measured by Surface Plasmon Resonance (SPR) on a Biacore T200 instrument using Protein-A coated S series chips. The target antibodies were individually captured for 60 sec at 30 μL/min to a level of 1000-2000RU onto this surface. The epitope scaffolds were subsequently injected as analytes at five concentrations using the single cycle injection method. The association phase was carried out for 180 seconds and the dissociation was done for 900 seconds with HBS-EP+ buffer flowing at 30 μL/min. Regeneration of the binding surface was done in 10 mM glycine-HC, pH=2 for 30 seconds at 30 μL/min with a 30 second baseline stabilization. A 1:1 Langmuir or Heterogeneous ligand model was used for data fitting and analysis. For $K_D$s between Spike proteins and target antibodies, neutravidin (Thermo Scientific) was first captured for 30-60 sec at 30 μL/min to a level of 1000–2000RU using CM5 S-series chips. Then biotinylated Spike proteins (WA-2, 2 P, R&D systems) were subsequently captured to a level of 3-500RU by the neutravidin. FAB fragments of antibodies were then injected as analytes at five concentrations using the single cycle method described above.

**Antibody binding by ELISA.** Spike$^{815–823}$ peptide (residues 808-833) and stem helix (1140-1163) peptides and mutated variants were commercially synthesized (Genscript). The following Spike proteins were used: WA-2, 2 P; SARS-CoV-1, 2 P; MERS, 2 P; OC43, 2 P; HKU-1, 2 P; GXP4L, 2 P; SHC014, 2 P; RaTG13, 2 P, (all produced according to[32,33]); and Delta; BA1.1, Hexapro; XBB, Hexapro; 229E; NL63; (all from Sino Biological). Peptides and Spikes were diluted to 2 μg/mL in 0.1 M Sodium Bicarbonate and epitope scaffolds were diluted to 100 μg/mL. Antigens were coated onto 96 or 384 well high binding enzyme-linked immunosorbent assay (ELISA) plates (Corning) by overnight incubation at 4 C. Plates were washed with Superwash buffer (1X PBS supplemented with 0.1% Tween-20) and blocked for 1 hour at room temperature with Superblock buffer supplemented with azide (80 g Whey Protein, 300 mL Goat Serum, 20 mL aqueous 5% Sodium Azide, 10 mL Tween20, 80 mL of 25X PBS, diluted to 2 L). Antibodies starting at 100 μg/mL were serially diluted in Superblock with azide using a threefold serial dilution, added to the plates, and incubated at room temperature for 1 hour. Plates were then washed twice with Superwash and goat anti Human IgG-HRP secondary antibody (Jackson ImmunoResearch Laboratories; Code Number: 109-035-098; Lot number: 154823) diluted 1:15,000 in SuperBlock (without Sodium azide) was subsequently added. After incubating at room temperature for 1 hour, the plates were washed four times in Superwash. Room temperature TMB (Tetramethylbenzidine) substrate was added and after 5 minutes elapsed, the reaction was stopped by acid stop solution (0.33 N HCl). Absorption was measured at 405 nm on a Cytation 1 plate reader (BioTek). Data was analyzed and plotted with Prism version 10.0.0 (Graphpad).

**Sera binding by ELISA.** Antigens were diluted to 2 μg/mL in 0.1 M Sodium Bicarbonate and coated onto 384 well high binding enzyme-

linked immunosorbent assay (ELISA) plates (Corning) by overnight incubation at 4 C. Plates were washed with Superwash buffer (1X PBS supplemented with 0.1% Tween-20) and blocked for 1 hour at room temperature with Superblock buffer supplemented with azide (80 g Whey Protein, 300 mL Goat Serum, 20 mL aqueous 5% Sodium Azide, 10 mL Tween20, 80 mL of 25X PBS, diluted to 2 L). Mouse sera starting at a dilution ratio of 1:50 and control antibodies starting at 100 μg/mL were serially diluted in Superblock with azide using a fivefold serial dilution scheme. Human sera starting at a dilution ratio of 1:30 and control antibodies starting at 100 μg/mL were serially diluted in Superblock with azide using a threefold serial dilution scheme. Sera and controls were added to the plates and incubated at room temperature for 1.5 hours. Plates were then washed twice with Superwash and the corresponding IgG-HRP secondary antibody (goat anti mouse IgG-HRP, 1:10,000 dilution, Jackson ImmunoResearch Laboratories; Code Number: 115-035-071; Lot number: 158206; or goat anti human IgG-HRP, 1:15,000 dilution, Jackson ImmunoResearch Laboratories, Code Number: 109-035-098; Lot number: 154823) in SuperBlock (without Sodium azide) was subsequently added. After incubating at room temperature for 1 hour, the plates were washed four times in Superwash. Room temperature TMB (Tetramethylbenzidine) substrate was added and after 15 minutes elapsed, the reaction was stopped by acid stop solution (0.33 N HCl). Absorption was measured at 405 nm on a SpectraMax Plus plate reader (Molecular Devices). Data was analyzed and plotted with Prism version 10.0.0 (Graphpad).

## Structural analysis by x-ray crystallography

Crystallography experiments were performed using the sitting drop vapor diffusion technique. 96-well crystallization plates were set, in which the protein of interest was mixed in a 1:1 ratio with reservoir solution. For S2hlx-Ex19, crystals were obtained when a 9.5 mg/mL sample was mixed with 0.1 M citric acid, 3.0 M sodium chloride, pH 3.5 (Index Screen) at 4 °C. S2hlx-7 and antibody DH1057.1 were mixed in 1:1 molar ratio (300μM) and incubated for one hour at 4 C. Crystals for the complex were obtained in 25% PEG 3350 at room temperature. FP-15 and antibody DH1058 were mixed in 1:1 molar ratio (60μM) and incubated for one hour at 4 C. Crystals for the complex were obtained in 0.2 M sodium malonate pH 7 and 25% PEG 3350 at room temperature. Prior to data collection, crystals were cryoprotected in mother liquor supplemented with 20% glycerol before being plunge-frozen into liquid nitrogen. Diffraction data for S2hlx-Ex19 and FP15-DH1058 were collected at APS SER-CAT 22-ID-D and diffraction data for S2hlx-7-DH1057.1 were collected at APS SER-CAT 22-BM-D. All datasets were collected at cryogenic conditions with a wavelength of 1.00 Å. A full description of the crystallographic data collection and refinement statistics can be found in Supplementary Table 1. Diffraction data were indexed in iMOSFLM[34] and scaled in AIMLESS[35]. Molecular replacement solutions for both S2hlx-Ex19 and the S2hlx-7-DH1057.1 complex were found in PhaserMR[36], using PDB ID: 3N1B as a search ensemble for S2hlx-Ex19, PDB ID: 3LMO as a search ensemble for S2hlx-7 and PDB ID: 6UOE as a search ensemble for DH1057.1. Coordinates for S2hlx-Ex19 and S2hlx-7-DH1057.1 were iteratively built and refined using Coot[37], ISOLDE[38] and Phenix[39] to Rwork/Rfree values of 25.4/ 27.5 and 21.1/24.5, respectively. The presence of translational non-crystallographic symmetry in the S2hlx-7-DH1057.1 crystal likely accounts for its relatively high Rfree value and the L-test did not indicate the presence of any twinning. Figures showing structures were prepared in Pymol v2.5.5 (Schroedinger).

## Reactivity of designed epitope scaffolds with human samples

Human subject studies were approved by the Duke University Health System Institutional Review Board (IRB) and conducted in agreement with the policies and protocols approved by the Duke IRB, consistent with the Declaration of Helsinki. Written informed consent was obtained from all research subjects or their legally authorized representatives. Study participants with SARS-CoV-2 acute infection were enrolled in the Molecular and Epidemiological Study of Suspected Infection protocol (MESSI, IRB Pro00100241) at Duke University, and were followed longitudinally. Samples were selected for this study from participants who had seroconverted and had symptom onset more than 10 days prior. From patient 26 and patient 29, PBMCs were collected and analyzed by FACS at the same time point used for sera analysis for patient 26 and 7 days later for patient 29. Samples from vaccinated only participants were obtained from subjects enrolled in the Study of Immune Response to COVID-19 Vaccines (IRB Pro00107929) at Duke University. Supplementary Table 2 describes the vaccination protocol, SARS-CoV-2 variant detected and the time of collection for the samples used in this analysis.

Human sera were analyzed as described above in *sera binding by ELISA* method. Statistical differences were tested using the Wilcoxon signed rank test.

## Isolation and analysis of epitope scaffold reactive B cells

We prepared B cell tetramers of biotinylated S2hlx-Ex19 or FP-10 by mixing with Streptavidin-VB515 (Miltenyi Biotec) and Streptavidin-AlexaFluor 647 (ThermoFisher Scientific) at 4:1 molar ratio. A scaffold-only tetramer was also made with biotinylated S2hlx-Ex19-PShlx or FP-10-wt mixed with Streptavidin-BV421 (Biolegend). These probes did not contain the grafted epitope.

Cryopreserved PBMCs were thawed in warm RPMI-1640 containing 10% FBS, then counted. Cells were stained with pre-optimized concentrations of the following antibodies: PE anti-human IgD (clone IA6-2, BD Biosciences, 1:300 final concentration), PE-TXRD anti-human CD10 (clone HI10A, BD Biosciences, 1:300), PE-Cy5 anti-human CD3 (clone HIT3a, BD Biosciences, 1:40), PE-Cy7 anti-human C27 (clone O323, ThermoFisher Scientific, 1:150), AlexaFluor 700 anti-human CD38 (clone LS198-4-3, Beckman Coulter, 1:40), APC-Cy7 anti-human CD19 (clone SJ25C1, BD Biosciences, 1:80), BV570 anti-human CD16 (clone 3G8, Biolegend, 1:40), BV605 anti-human CD14 (clone M5E2, Biolegend, 1:40), and BV711 anti-human IgM (clone G20-127, BD Biosciences, 1:160). S2hlx-Ex19-VB515- and -AF647, and S2hlx-Ex19-PShlx-BV421 were added to identify stem helix epitope-specific B cells. Cells were incubated with Aqua Live/Dead (ThermoFisher Scientific, 1:1000) to exclude dead cells. Cells were acquired on a BD S6 cell sorter (BD Biosciences). Data were analyzed using FlowJo v10.8 (BD Biosciences).

Immunoglobulin heavy and light chain variable regions (VH and VK/L) from singly sorted antigen-specific memory B cells were RT-PCR amplified using SuperScript III and AmpliTaq Gold 360 Master Mix (Thermo Fisher Scientific, Waltham, MA) under conditions previously described[22]. PCR products were purified in Biomek FX Laboratory Automation Workstation (Beckman Coulter, Indianapolis, IN) and sequenced by Sanger sequencing. The V(D)J rearrangement, somatic hypermutation frequency, CDR3 length of VH and VK/L chains, and antibody clonal lineages were analyzed using the software Cloanalyst[40]. The heavy chains of mAbs DH1493, DH1501, and DH1502 were assigned to the same clone by the software package Cloanalyst. Unmutated common ancestor (UCA) inference was performed using the heavy and kappa chain pairs of the three mAbs using the paired-chain inference implementation of the UCA inference part of the software package Cloanalyst.

RT-PCR amplified sequences were transiently expressed as previously described[22]. Briefly, the linear expression cassettes were constructed by overlapping PCR to place the PCR-amplified VH and VK/L chain genes under the control of a CMV promoter along with heavy chain IgG1 constant region or light chain constant region and a BGH ploy A signal sequence. The linear expression cassettes of heavy and light chains were then co-transfected into 293 T cells in 6-well plates. After 3 days, the cell culture supernatants were harvested and concentrated for binding assays. For antibodies of interest, recombinant IgG1 monoclonals were expressed and purified as described above.

## Pseudovirus neutralization assay

The pseudovirus neutralization assay performed at Duke has been described in detail[41] and is a formally validated adaptation of the assay utilized by the Vaccine Research Center; the Duke assay is FDA approved for D614G and other SARS-CoV-2 variants. For measurements of neutralization, pseudovirus was incubated with 8 serial 5-fold dilutions of antibody samples in duplicate in a total volume of 150 μl for 1 hr at 37 C in 96-well flat-bottom culture plates. 293 T/ACE2-MF cells (obtained from Drs. Mike Farzan and Huihui Mu at The Scripps Research Institute) were detached from T75 culture flasks using TrypLE Select Enzyme solution, suspended in growth medium (100,000 cells/ml) and immediately added to all wells (10,000 cells in 100 μL of growth medium per well). One set of 8 wells received cells + virus (virus control) and another set of 8 wells received cells only (background control). After 71–73 hrs of incubation, medium was removed by gentle aspiration and 30 μl of Promega 1X lysis buffer was added to all wells. After a 10-minute incubation at room temperature, 100 μl of Bright-Glo luciferase reagent was added to all wells. After 1-2 minutes, 110 μl of the cell lysate was transferred to a black/white plate. Luminescence was measured using a GloMax Navigator luminometer (Promega). Neutralization titers are the inhibitory dilution (ID) of serum samples at which RLUs were reduced by 50% (ID50) compared to virus control wells after subtraction of background RLUs. Serum samples were heat-inactivated for 30 minutes at 56 C prior to assay.

## Infected-cell antibody binding assay

The binding of isolated anti-SARS-CoV-2 monoclonal antibodies to infected cells was measured as previously reported[42]. Briefly, Vero E6 cells (ATCC CRL-1587) expressing TMPRSS2 and ACE2 and infected with either D614G (Germany/BavPat1/2020) or BA.1 (hCoV-19/USA/MD-HP20874/2021) variants were incubated with TrypLE Select (Gibco) for 15 minutes at 37 °C to detach cells and washed with PBS. Monoclonal antibodies were then added to infected cells at 2, 10 or 50 μg/mL. Approximately $2\times10^5$ infected cells were incubated with the mAbs for 30 minutes at room temperature, washed and then incubated with vital dye (Live/Dead Far Red Dead Cell Stain, Invitrogen) for 15 minutes at room temperature to exclude nonviable cells from subsequent analysis. Cells were then washed with Wash Buffer (1%FBS-PBS; WB), pelleted by centrifugation and incubated with 1 mL of 4% Methanol-free Formaldehyde (Duke GHRB SOP 38; Attachment 17) for 30 minutes at room temperature. Cells were then washed twice with Wash Buffer, permeabilized with CytoFix/CytoPerm (BD Biosciences) and stained with A568-conjugated anti-SARS-CoV-2 nucleocapsid antibody (1 μg/mL; 40143-MM08, Sino Biological) and PE/Cy7-conjugated secondary anti-Human IgG Fc antibody (10 μL/mL, Clone: HP6017, Biolegend) for 30 minutes at room temperature. Cells were washed and resuspended in 250 μL PBS–1% paraformaldehyde. Samples were acquired within 24 h using a BD Fortessa cytometer and a High Throughput Sampler (HTS, BD Biosciences). Data analysis was performed using FlowJo 10 software (BD Biosciences). Gates were set to include singlet, live, nucleocapsid+ (NC + ) and IgG+ events. Binding to mock infected cells was measured using the live cell gate as there were no NC+ events. All final data represent specific binding, determined by subtraction of non-specific binding observed in assays performed with mock-infected cells.

## Antibody-dependent NK cell degranulation assays (infected and spike-transfected)

Cell-surface expression of CD107a was used as a marker for NK cell degranulation and performed as previously described[42]. Briefly, target cells were 293 T (ATCC, CRL-3216) cells 2-days post transfection with a SARS-CoV-2 S protein (D614G) expression plasmid. Natural killer cells purified by negative selection (Miltenyi Biotech) from peripheral blood mononuclear cells obtained by leukapheresis from a healthy, SARS-CoV-2-seronegative individual (Fc-gamma-receptor IIIA (Fcγ-RIIIA

158 V/F heterozygous) previously assessed for Fcγ-RIIIA genotype and frequency of NK cells were used as a source of effector cells. NK cells were incubated with target cells at a 1:1 ratio in the presence of monoclonal antibodies, Brefeldin A (GolgiPlug, 1 μl/ml, BD Biosciences), monensin (GolgiStop, 4 μl/6 mL, BD Biosciences), and anti-CD107a-FITC (25 μL/mL BD Biosciences, clone H4A3) in 96-well flat bottom plates for 6 hours at 37 °C and 5% $CO_2$. NK cells were removed from the wells and stained for viability prior to staining with CD56-PECy7 (3.125 μL/mL, BD Biosciences, clone NCAM16.2), CD16-PacBlue (12.5 μL/mL, BD Biosciences, clone 3G8), and CD69-BV785 (6 μL/mL, Biolegend, Clone FN50). After three washes, cells were resuspended in 115 μL PBS–1% paraformaldehyde. Flow cytometry data analysis was performed using FlowJo software (v10). Data is reported as the area under the curve (AUC) of %CD107a+ live NK cells (gates included singlets, lymphocytes, aqua blue-, CD56+ and/or CD16 + , CD107a + ), calculated as previously described[42] at three concentrations of monoclonal antibody: 2, 10 and 50 μg/mL. All final data represent specific activity, determined by subtraction of non-specific activity observed in assays performed with mock-infected cells and in the absence of antibodies, and in the presence of a non-specific monoclonal antibody, the anti-HIV-1 antibody VRC01.

## Development of mi03 nanoparticles conjugated with epitope scaffolds for immunizations

**SpyCatcher003-Mi3 Nanoparticles.** Plasmid encoding for the spycatcher003-mi3 particles was acquired from Addgene (Plasmid #159995) and transformed into *E. coli* BL-21 (RIPL) (Agilent Technologies) cells. Five milliliters starter cultures were grown overnight at 37 C in Lysogeny Broth (LB) supplemented with 1% glucose and 50 μg/mL kanamycin. Cultures were diluted 1:100 in Terrific Broth (TB) media (RPI) supplemented with kanamycin and grown at 37 C to an $OD_{600}$ of ~0.6. The temperature was subsequently lowered to 20 C, and the cultures were grown to $OD_{600}$ of ~0.8 and induced with isopropyl β-D-1-thiogalactopyranoside (IPTG) to a final concentration of 0.5 mM. The cultures were shaken for 16–18 hours at 250 rpm. Cell pellets were collected by centrifugation at 6,000xg and the pellets were stored overnight at -20C. Pellets were resuspended in 40 mL of 25 mM Tris-HCl, 300 mM NaCl, pH=8.5 supplemented with a working concentration of 2 mM PMSF dissolved in isopropanol, 40 mg lysozyme (0.1 mg/mL) and 1 tablet of protease inhibitor. The lysates were incubated at room temperature for 30 minutes on a platform shaker and later sonicated for 13 minutes with 10 s on and 30 s off 50% duty-cycle. The sonicated lysates were centrifuged for 45 minutes at 18,000xg at 4 C. To precipitate the spycatcher003-mi3 nanoparticles, 170 mg/mL of ammonium sulfate was added to the supernatant solution and subsequently shaken at 4 C for 1 hour. Precipitated particles were collected by centrifuging at 16,000 × g for 30 minutes at 4 C. The supernatant was discarded, and the pellet was washed with endotoxin-free water to remove residual salty buffer. The nanoparticles were resolubilized in 8 mL of 25 mM Tris-HCl, 150 mM NaCl, pH=8.5. To remove insoluble aggregates, the solubilized nanoparticles were centrifuged at 18,000 × g for 30 minutes. Nanoparticle expression and purity was confirmed by SDS-PAGE analysis, and quantified spectrophotochemically at 280 nm on a Nanodrop 2000 (ThermoFisher Scientific). The spycatcher003-mi3 nanoparticles were concentrated in a 10 kDa MW spin columns (ThermoFisher Scientific) and further purified by size exclusion chromatography in 25 mM Tris, 150 mM NaCl, pH=8.0, on an AKTA-Go FPLC (Cytiva) using a Superose 6 Increase 10/300GL column. Fractions containing the spycatcher003-mi3 nanoparticles were pooled and concentrated as above. The nanoparticles were stored at -80C.

**Conjugation of epitope scaffolds to Spycatcher003-mi3 nanoparticles.** Epitope scaffolds were mixed with spycatcher003-mi3 nanoparticles in a 1.2:1 molar ratio. The complex was mixed

thoroughly and incubated overnight on ice at 4 C. Conjugation was confirmed by SDS-PAGE. The conjugated nanoparticles were separated from excess monomeric epitope scaffolds by size-exclusion chromatography in 25 mM Tris, 150 mM NaCl, pH 8.0, on an AKTA-Go FPLC (Cytiva) using a Superose 6 Increase 10/300GL column. Fractions containing the epitope-scaffold-mi3 nanoparticles were pooled and concentrated as above. Endotoxins were removed by treating the nanoparticles with 2% triton X-100 as previously described[43]. Endotoxin levels were tested by a chromogenic endotoxin quantification kit (ThermoFisher Scientific). The endotoxin-free nanoparticles were stored at -80C.

Nanoparticles were analyzed by Negative Stain Electron Microscopy to confirm that they were well formed. A frozen aliquot was thawed in RT water bath, then placed on ice. The sample was then diluted to 400 µg/ml with 5 g/dl Glycerol in HBS (20 mM HEPES, 150 mM NaCl pH 7.4) buffer containing 8 mM glutaraldehyde. After 5 min incubation, glutaraldehyde was quenched by adding sufficient 1 M Tris stock, pH 7.4, to give 80 mM final Tris concentration and incubated for 5 min. Quenched sample was applied to a glow-discharged carbon-coated EM grid for 10-12 second, then blotted, and stained with 2 g/dL uranyl formate for 1 min, blotted and air-dried. Grids were examined on a Philips EM420 electron microscope operating at 120 kV and nominal magnification of 49,000x, and 30 images were collected on a 76 Mpix CCD camera at 2.4 Å/pixel. Images were analyzed by 2D and 3D class averages using standard protocols with Relion 3.0[44].

### Mouse immunization and protection studies

**Immunization studies in BALB/c mice.** BALB/c (#028) female mice were purchased from Charles River. All mouse studies were performed under an approved Duke University IACUC protocol. All animal rooms were kept on a 12/12 light cycle unless otherwise requested. Heat and humidity were maintained within the parameters outlined in The Guide for the Care and Use of Laboratory Animals and animals were fed a standard rodent diet. Mice were housed in individually ventilated micro-isolator caging on corn cob bedding. The Toll-like receptor 4 agonist glucopyranosyl lipid adjuvant–stable emulsion (GLA-SE) was used as the adjuvant for the vaccine immunogens. Groups of mice ($n = 8$) were vaccinated intramuscularly with GLA-SE-adjuvanted Ex15-NP; GLA-SE-adjuvanted Ex_mosaic-NP; GLA-SE-adjuvanted mixture of equal parts Ex15-NP, Ex17_NP, Ex19_NP and Ex20_NP; or GLA-SE-adjuvanted Ex15-NP (prime), followed by Ex19_NP (boost 1) and Ex20_NP (boost 2). An adjuvant-only group ($n = 4$) was included for comparison. Vaccine immunogens were administered at 5 µg and formulated with 5 µg of adjuvant. Mice were immunized at week 0, week 4, and week 8. Blood samples were collected either 7 days prior to immunization (pre-bleed), or 7 days after each immunization. Seven days after the final dose mice were sacrificed for terminal analysis.

**Live virus protection studies.** Nineteen to twenty-one female k18-hACE2 mice purchased from Jackson Laboratory (B6.Cg-Tg(K18-ACE2)2 Prlmn/J; JAX strain number #034860) were used for the WIV-1 viral challenge studies. The Toll-like receptor 4 agonist glucopyranosyl lipid adjuvant–stable emulsion (GLA-SE) was used as the adjuvant for the vaccine immunogens. Mouse vaccination studies were performed intramuscularly twice with mRNA ACLNP 307 2019NCOV WUHAN S-2P spike followed by two additional shots of either mRNA spike or GLA-SE-adjuvanted Ex_mosaic-NP. Protein immunogens were administered at 10 µg formulated with 5 µg of adjuvant. mRNA immunogens were administered at 20 µg. Mice were immunized at week 0, week 4, week 8, and week 12. Blood samples were collected either 7 days prior to immunization (pre-bleed), or 7 days after each immunization. Mice were then moved into the BSL3 and acclimated. For infection, vaccinated mice were anaesthetized with ketamine/xylazine and infected

with $1 \times 10^4$ PFU WIV1-CoV intranasally in a total volume of 50ul[45]. Infected mice were weighed daily.

Mouse studies were performed according to the recommendations for the care and use of animals by the Office of Laboratory Animal Welfare (OLAW), National Institutes of Health, and the Institutional Animal Care and Use Committee (IACUC) at the University of North Carolina (UNC permit no. A-3410-01). All infectious work was performed in Biosafety Level 3 laboratories (BSL-3) with approved standard operating procedures and safety conditions for SARS-CoV-2. All animal rooms were kept on a 12/12 light cycle unless otherwise requested. Temperature was maintained between 20-23.3 C and humidity was kept at 30-70%. Animals were fed a standard rodent diet (PicoLab Select Rodent 50 IF/6 F - 5V5R). Mice were housed in individually ventilated micro-isolator caging on corn cob bedding.

For virus titration by plaque assay, the caudal lobe of the right lung was homogenized in phosphate-buffered saline (PBS), and the homogenate was serial diluted and inoculated onto Vero E6 cells (American Type Culture Collection (ATCC), CRL1586), followed by agarose overlay. Plaques were visualized with an overlay of neutral red dye on day 2 after infection. At indicated time points, mice were euthanized and gross pathology (congestion score) of the lung tissue was assessed and scored on a scale from 0 (no lung congestion) to 4 (severe congestion affecting all lung lobes).

### Statistics and reproducibility

The statistical analyses performed are described in Methods and figure legends. The Investigators were not blinded to the selection of the human specimens used for sera analysis. Samples were preferentially selected from participants who had seroconverted and had symptom onset more than 10 days prior to collection. No statistical method was used to predetermine the number of samples analyzed in the vaccinated, infected and vaccinated and infected groups. No statistical method was used to predetermine the number of animals in a group for the animal studies. No collected data were excluded from the analyses.

### Reporting summary

Further information on research design is available in the Nature Portfolio Reporting Summary linked to this article.

## Data availability

All relevant data supporting the key findings of this study are available within the article and its Supplementary Information files. Crystal structures have been deposited in the PDB under accession numbers 8F5I, 8FDO, and 8F5H. Plasmids encoding the engineered epitope scaffolds can be obtained through an MTA from the corresponding author. Source data are provided with this paper.

## Code availability

The design of the scaffolds described herein was undertaken using publicly available RosettaScripts v3.9 within the Rosetta framework (https://www.rosettacommons.org/). Specific RosettaScripts are described in detail in the method section and publicly available (https://github.com/AzoiteiLab/S2-scaffold-scripts)[46]. Further optimization and in silico analysis was undertaken using ProteinMPNN and Alphafold v2.1 (https://github.com/google-deepmind/alphafold) both of which are freely available. Specific scripts and settings used for ProteinMPNN and Alphafold analysis are described in the methods. Scripts can be found at https://github.com/AzoiteiLab/S2-scaffold-scripts.

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

## Acknowledgements

We thank Whitney Edwards Beck, Deborah Murray, Micah McClain, Thomas Burke, Jack Anderson, Lynn Harrington, Christopher Todd and

Kathlene Chmielewski for help with acquiring and managing the human samples. We thank Aria Arus-Altuz for developing the fluorescent probes used for FACS. We thank Brian Watts, Ken Cronin and Alam Munir for their assistance at the DHVI Biomolecular Interaction Analysis (BIA) Core Facility. We thank Wes Rountree and Yunfei Wang for the statistical analysis. We thank Advaita Sing and Christopher O'Donnell from the Azoitei lab for their help. We thank Jacob Gater and Pahvie Chhan from the Duke Human Vaccine Institute Small Animal Team for their help with the immunization studies. We thank Priyamvada Acharya and Jared Lindenberger for their help with crystallization experiments. Structural results shown in this report are derived from data collected at Southeast Regional Collaborative Access Team (SER-CAT; 22 ID and 19 BM) beamlines at the Advanced Photon Source, Argonne National Laboratory. Use of the Advanced Photon Source was supported by the U. S. Department of Energy, Office of Science, Office of Basic Energy Sciences, under Contract No. W-31-109-Eng-38. This work was supported by grants P01AI158571 (B.F.H) and R01AI155804 (M.L.A) from the National Institute of Allergy and Infectious Diseases of the National Institute of Health and by a Duke School of Medicine COVID Award (M.L.A).

## Author contributions

M.L.A., A.B.K., and D.J.M.: conceptualization, methodology, study design, and immunogen engineering; A.B.K., C.H., C.B., Ka.W., D.J.M., B.R. and P.V. performed in vitro characterization of epitope scaffold immunogens; A.B.K., D.W., and Ka.W. performed the crystallographic analysis; Q.Y., L.X., A.F., R.P., M.B., K.O.S., and D.W.C. performed the B cell isolation, sequencing and antibody identification; A.E. and D.C.M. performed the pseudovirus neutralization assay; D.M., B.D., T.K., S.S.O., and G.F. performed NK degranulation analysis; A.N., C.H., A.S., J.M.F., A.H., N.J.C., M.L.M. and M.D.M. performed the animal immunization, the viral challenge experiments and their analysis; C.W., E.A.P., E.B.W., Ke.W., K.O.S., R.B., and B.F.H.: resources; Ke.W. and E.V.I. preformed unmutated common ancestor interference; K.M. and R.J.E. performed NSEM analysis; figure development; M.L.A., D.J.M., A.B.K.: writing, original draft; M.L.A., D.J.M., A.B.K., B.F.H., D.W., Ke.W., K.O.S., D.M., G.F., A.E., D.C.M., A.N., A.S., L.X., E.V.I.: writing, review and editing; M.L.A., B.F.H., Ke.W., K.O.S.: funding acquisition; M.L.A.: project administration and supervision.

## Competing interests

M.L.A., B.F.H., D.J.M. and B.K. submitted a patent application that covers the engineered epitope scaffolds. The other authors declare no competing interests.

## Additional information

[1]Duke Human Vaccine Institute, Duke University, Durham, NC, USA. [2]Department of Medicine, Duke University, Durham, NC, USA. [3]Department of Surgery, Duke University, Durham, NC, USA. [4]Center for Human Systems Immunology, Duke University, Durham, NC, USA. [5]Department of Epidemiology, University of North Carolina at Chapel Hill, Chapel Hill, NC, USA. [6]Center for Infectious Diseases and Diagnostic Innovation, Duke University Medical Center, Durham, NC, USA. [7]Department of Pediatrics, Duke University, Durham, NC, USA. [8]Department of Molecular Genetics and Microbiology, Duke University, Durham, NC, USA. [9]Department of Immunology, Duke University, Durham, NC, USA. [10]These authors contributed equally: A. Brenda Kapingidza, Daniel J. Marston. ✉e-mail: mihai.azoitei@duke.edu

