## [Peer Review File · Nature Communications]

Engineered Immunogens to Elicit Antibodies Against Conserved Coronavirus EpitopesReviewer #1 (Remarks to the Author):

In this manuscript from Kapingidza et al the authors design, produce, and characterize epitope scaffolds that display epitopes of SARS-CoV-2 S2-directed antibodies that target the fusion peptide and stem helix. These epitopes are highly conserved, and the antibodies that recognize them bind broadly to SARS-CoV-2 spike variants as well as spikes from other betacoronaviruses. Elicitation of such antibodies may be important for the development of broadly protective coronavirus vaccines, and the epitope scaffolds developed here are intended for use as vaccine antigens. The authors succeed in developing epitope scaffolds that bind with high affinity to representative S2-directed antibodies, and high-resolution crystal structures demonstrate that the epitope scaffolds adopt conformations that are very close to the designed models. These scaffolds react with human sera and memory B cells obtained from people that had been infected with and or/vaccinated against SARS-CoV-2.

The manuscript is well written, although it can be difficult to keep track of all the various scaffold names. The biophysical binding studies were performed well, as were the X-ray crystallography studies, as evidenced by the very good X-ray data collection and refinement statistics. The major disappointment is the lack of immunogenicity studies, which are standard in the field when developing novel immunogens. Without immunogenicity data, the importance of these immunogens (and the manuscript itself) cannot be ascertained.

Other comments:

1. lines 90 & 144: "dissociation constants" should be "equilibrium dissociation constants"
2. lines 105 & 106: A better rationale is needed for these mutagenesis studies. The "affinity gains" could be due to a combination of antibody contacts with the epitope and the scaffold. So mutation of the epitope residues does not rule out that additional contacts with the scaffold contribute to the affinity.
3. Line 115 and elsewhere: when RMSDs are listed, the atoms involved in the calculation need to be provided. Were these based on all atoms in the residue, mainchain, Calpha?
4. The discussion is not thorough, and does not put the results of the study into the context of prior literature related to immunogen design and epitope scaffolds for SARS-CoV-2 and related viruses.

Reviewer #2 (Remarks to the Author):

Summary:

The authors apply structure-based design and computational modeling to engineer epitope scaffolds to display conserved vaccine targets from the relatively invariant S2 submit of SARS-CoV-2 spike protein. This scaffolding approach includes presentation of the fusion peptide and stem helix epitopes. The authors evaluate antigenicity of the scaffolded epitopes by testing binding with known monoclonal antibodies against these sites. The authors report pM to nM binding affinity, and in some cases show that the inferred antibody germ lines engage the scaffolded epitopes, namely those of the stem helix. Their evaluation also includes high-resolution crystal structures of mature antibodies in complex with the scaffolded epitopes. The authors then apply their immunogens to assay for reactivity in patient immune sera and also as a B cell flow cytometry probes where the authors report reactivity against memory B cells from human PBMC.

Overall:

Enthusiasm is high for the design and precision grafting/scaffolding of these important vaccine target epitopes. This aspect of the study is non-trivial, innovative, rigorous, comprehensive and aside from my request to include a key positive control in their antigenicity studies, the work is beautiful! However, downstream immunological testing of their scaffolded products is weak and

superficial. The reviewer feels that publication in this journal necessitates addressing these deficits, otherwise the work stands as a case of outstanding protein engineering but with largely untested activity in the immunological/vaccine space, which is after all, the motivation behind the study.

Questions:

- 1) Figure 1e,f, S4. The authors should include a measure of mab binding to the native conformational S-trimer to experimentally confirm the affinity gains in their system (not just cite literature values).
- 2) Figure 1. What about binding to UCAs from DH1058 and DH1294? (And comparison to native S trimer therein.)
- 3) Figure 2b,c. Again the authors should include a measured comparison to native S trimer. This would be particularly informative in understanding whether the UCA binding is really a 'gain of function' due to the scaffolding approach.
- 4) Figure 3e. As above, include measured comparison to S trimer.
- 5) Figure 4d, S18. The gating for B cell memory from human PBMC is underdeveloped. The authors need to include CD27 as marker for memory (IgD-/CD27-). Also in humans, CD38 expression on memory B cells is more intermediate (+/-): the CD38 high population that the authors include in the memory gate will be plasmablasts (CD38++). Authors should consult a recent review on this subject (e.g. PMID: 31681331)
- 6) Figure 4. The immunological readouts for these scaffolded epitopes is very underdeveloped. It is good, but not surprising that there is reactivity against these sites either in the immune sera or in at the antigen specific B cell level. The authors should fix their gating scheme for B cell memory and then clone out the BCRs and demonstrate biochemically that they have the appropriate target affinity for these sites on S2 (affinity measurements, competition assays).
- 7) An overall issue is that the authors are present an immunogen paper without properly testing a preclinical vaccine parameter. Demonstration of preclinical activity is not beyond the scope of an immunogen design paper (and should be the goal!), particularly if the study is sent to a higher impact journal such as the case here. What happens if you prime mice with S trimer (as an introduction of pre-immunity) and then boost with the engineered scaffolds? Do you boost responses/expand B cell memory against the fusion peptide or stem helix epitopes? Is there an strong anti-scaffold response?
An argument could be made that mouse antibody repertoires are less appropriate given the reactivity to human UCAs; however, if that is the case then there are still straightforward metrics to test in the preclinical space. Namely, when the immunogens are applied as B cell flow probes to PBMC, do they enrich for S2 bnAb precursors from the human germline B cell repertoire (IgD+/IgM+/IgG-/CD27-). This approach has become pretty standard in the germline targeting vaccine space (e.g. HIV, flu and in SARS-CoV-2). In the case of human S2 broadly neutralizing antibodies, non-random patterns in VH/VL usage have been described (e.g. heavy bias in IGHV1-46 and IGKV3-20 usage in bnAbs targeting the stem helix, PMID: 35291291). A clear prediction is to test whether the authors epitope scaffold immunogens actually enrich for bnAbs precursors with similar VH/VL biases.

Reviewer #3 (Remarks to the Author):

The authors of this manuscript applied state-of-the-art computational protein design methods to design a range of proteins that present two epitopes of the coronavirus spike protein. They used ELISA and surface plasmon resonance (SPR) to show that the designed proteins have similar or higher affinities to the native peptides in binding to antibodies. They showed that the binding

modes of the epitope-presenting proteins agree with the designed models by mutating the hotspot residues and solving x-ray crystal structures. In the end, they demonstrated that the designed proteins could bind to antibodies and B-cells in the blood of volunteers who have been previously exposed to SARS-CoV2. This manuscript describes an interesting biomedical application of computational protein design, which is a promising direction of the field. The methodology explored by this work and the designed products may potentially lead to novel vaccines. I suggest the journal accept this manuscript if the major issue can be solved.

Major issue

The dissociation constants of the designed proteins were determined by SPR experiments. However, the SPR data quality was poor. In Fig. 2c and supplementary figures, the SPR traces are noisy and far from converging. The kinetic constants fitted from these data are not reliable. For example, The fitted k_{on} of S2hlx-7 to S2P6 was 1.3×10^7 . But the binding curve of S2hlx-7 in Fig. 2c doesn't converge faster than S2hlx-4, whose fitted k_{on} was 10^4 . Therefore, higher-quality SPR data or alternative assays for measuring dissociation constants are needed to support the main conclusion of the manuscript.

Minor issues

1. The most potent designs to bind CC40.8 are derived from S2hlx-Ex2, in which the epitope helix is an extension of the scaffold's terminal helix. The crystal structure of S2hlx-Ex19 showed that the epitope adopts a canonical alpha helix conformation without antibody binding. Since the designs derived from the other scaffolds are not as potent, it is questionable whether the designs are able to present the kinked helix conformation of the epitope. It would be better if the authors could discuss the limitation of the design method.
2. Some of the kinetic parameters in Fig. 1c and Fig. 2b are marked as "not tested." Although they probably don't affect the main conclusion of the paper, it would be better if fill these values in the main figures.

We are grateful to the Reviewers for their comments. We appreciate the kind words about the scaffold designs, and their assessment that the work is “non-trivial, innovative and rigorous”, and that the designs “may potentially lead to novel vaccines”. However, two Reviewers commented on the need for additional *in vivo* validation of the designed immunogens in order to illustrate their utility and potential impact. Therefore, we addressed this with a number of different studies summarized in two additional main text figures, **Figures 5 and 6** as follows:

- In **Figure 5**, the engineered epitope scaffolds were used to isolate from people with pre-existing SARS-CoV-2 immunity multiple antibodies against the target epitopes that had broad coronavirus reactivity. Some of these monoclonal antibodies had the same V_H/V_L pairing as previously isolated antibodies, while others were new, demonstrating the potential of our immunogens to engage and boost a polyclonal response in people with pre-existing immunity, who currently make up the majority of the population
- In **Figure 6**, we test the engineered immunogens targeting the stem helix epitope in mice and showed that they elicit sera with broad reactivity against human and animal betacoronaviruses, including SARS-CoV-2 VoCs, SARS-CoV-1, but also MERS, OC43 and HKU-1. In a separate protection study, we showed that the stem helix epitope scaffolds boost responses against the target epitope in animals primed with SARS-CoV-2 spike mRNA, resulting in protection from a heterologous viral challenge with WIV-1.

Taken together, we believe that these results validate our engineered immunogens *in vivo* and support their continued development and characterization to boost responses against conserved S2 epitopes as part of a next-generation pan betacoronavirus vaccine.

We also added additional binding characterization of our immunogens to other target antibodies and strengthened the SPR measurements as requested by one Reviewer. Altogether, we made substantial additions and edits to answer the Reviewer’s comments resulting in a significantly improved manuscript.

We note that in the time since our submission, a new manuscript (Shi et. al., Nature, 2023) demonstrated that what we and the rest of the field were calling the fusion peptide domain of SARS-CoV-2 is actually not the *bona fide* fusion domain; the fusion domain is located further upstream and is not engaged by antibodies like DH1058 or VN01H1 that were labeled as “fusion peptide antibodies” based on the incorrect domain assignment. We updated the manuscript to reflect this new domain assignment. We replaced the *fusion peptide epitope scaffold* references with *spike⁸¹⁵⁻⁸²³ epitope scaffolds* and updated previous fusion peptide references to this new domain.

We responded to the Reviewer’s comments below:

Reviewer #1 (Remarks to the Author):

Other comments:

1. *lines 90 & 144: “dissociation constants” should be “equilibrium dissociation constants”*

We made the requested modifications.

2. *lines 105 & 106: A better rationale is needed for these mutagenesis studies. The “affinity gains” could be due to a combination of antibody contacts with the epitope and the scaffold. So mutation of the epitope residues does not rule out that additional contacts with the scaffold contribute to the affinity.*

We agree with the reviewer that mutations outside the epitope could contribute to binding affinity. However, if present, such interactions are likely weak. We modified the text to indicate that our mutational analysis indicates that “the epitope is the major site of antibody interaction” (line 129).

3. *Line 115 and elsewhere: when RMSDs are listed, the atoms involved in the calculation need to be provided. Were these based on all atoms in the residue, mainchain, Calpha?*

We clarified that the RMSD is calculated over the backbone region as requested.

4. *The discussion is not thorough, and does not put the results of the study into the context of prior literature related to immunogen design and epitope scaffolds for SARS-CoV-2 and related viruses.*

We expanded the Discussion to put the design of epitope scaffold in the context of previous immunogen designs with this methodology and more thoroughly discussed the implications of our results (lines 456-492).

Reviewer #2 (Remarks to the Author):

Questions:

1) *Figure 1e,f, S4. The authors should include a measure of mab binding to the native confirmational S-trimer to experimentally confirm the affinity gains in their system (not just cite literature values).*

We conducted these additional measurements, now included in figures 1c and S4

2) *Figure 1. What about binding to UCAs from DH1058 and DH1294? (And comparison to native S trimer therein.)*

We conducted these additional measurements, now included in figures 1d and S5b

3) *Figure 2b,c. Again the authors should include a measured comparison to native S trimer. This would be particularly informative in understanding whether the UCA binding is really a 'gain of function' due to the scaffolding approach.*

We conducted these additional measurements, now included in figures 2b,c S7,8. Given the greater breadth and higher affinities of the S2hlx-Ex scaffolds we focused our studies of germline reactivity on these rather than the S2hlx scaffolds. We conducted ELISA measurements (figures S17b) to directly compare binding of spike and S2hlx-Ex scaffolds to a panel germline antibodies.

4) *Figure 3e. As above, include measured comparison to S trimer.*

We conducted these additional measurements, now included in figure S8.

5) *Figure 4d, S18. The gating for B cell memory from human PBMC is underdeveloped. The authors need to include CD27 as marker for memory (IgD-/CD27-). Also in humans, CD38 expression on memory B cells is more intermediate (+/-): the CD38 high population that the authors include in the memory gate will be plasmablasts (CD38++). Authors should consult a recent review on this subject (e.g. PMID: 31681331)*

We included CD27 and CD38 staining in the B cell analysis panel (as described in the Methods), but we did not use them in our gating strategy. We consider the sorted cells to be IgD- B cells, which are mostly memory B cells (which can be both CD27+ and CD27-), but can also include plasmablasts (CD27+CD38+). To clarify this, in the text, we now refer to the analyzed B cells as IgD- B cells and note that this population includes both memory B cells and plasmablasts.

6) *Figure 4. The immunological readouts for these scaffolded epitopes is very underdeveloped. It is good, but not surprising that there is reactivity against these sites either in the immune sera or in at the antigen specific B cell level. The authors should fix their gating scheme for B cell memory and then clone out the BCRs and demonstrate biochemically that they have the appropriate target affinity for these sites on S2 (affinity measurements, competition assays).*

As mentioned above, we conducted these additional antibody isolation and characterization experiments, please see Figure 5.

7) *An overall issue is that the authors are present an immunogen paper without properly testing a preclinical vaccine parameter. Demonstration of preclinical activity is not beyond the scope of an immunogen design paper (and should be the goal!), particularly if the study is sent to a higher impact journal such as the case*

here. What happens if you prime mice with S trimer (as an introduction of pre-immunity) and then boost with the engineered scaffolds? Do you boost responses/expand B cell memory against the fusion peptide or stem helix epitopes? Is there an strong anti-scaffold response?

An argument could be made that mouse antibody repertoires are less appropriate given the reactivity to human UCAs; however, if that is the case then there are still straightforward metrics to test in the preclinical space. Namely, when the immunogens are applied as B cell flow probes to PBMC, do they enrich for S2 bnAb precursors from the human germline B cell repertoire (IgD+/IgM+/IgG-/CD27-). This approach has become pretty standard in the germline targeting vaccine space (e.g. HIV, flu and in SARS-CoV-2). In the case of human S2 broadly neutralizing antibodies, non-random patterns in VH/VL usage have been described (e.g. heavy bias in IGHV1-46 and IGKV3-20 usage in bnAbs targeting the stem helix, PMID: 35291291). A clear prediction is to test whether the authors epitope scaffold immunogens actually enrich for bnAbs precursors with similar VH/VL biases.

The revised manuscript includes both antibody isolation and animal studies to demonstrate the utility of our scaffolds, please see new Figures 5 and 6 as described above. We indeed isolated antibodies with the IGHV1-46/IGKV3-20 immunogenetics as well as other different ones and characterized their neutralization and antibody-mediated functions (Figure 5). Figure 6 describes results from testing the stem helix immunogens in animals both by themselves and as boosts to spike mRNA. Stem helix epitope scaffolds elicited sera with broad activity and protected as boosts to mRNA against a challenge with a heterologous coronavirus, WIV-1.

Reviewer #3 (Remarks to the Author):

Major issue

The dissociation constants of the designed proteins were determined by SPR experiments. However, the SPR data quality was poor. In Fig. 2c and supplementary figures, the SPR traces are noisy and far from converging. The kinetic constants fitted from these data are not reliable. For example, The fitted k_{on} of S2hlx-7 to S2P6 was 1.3×10^7 . But the binding curve of S2hlx-7 in Fig. 2c doesn't converge faster than S2hlx-4, whose fitted k_{on} was 10^4 . Therefore, higher-quality SPR data or alternative assays for measuring dissociation constants are needed to support the main conclusion of the manuscript.

Based on the Reviewer's comments we realized that it was no clear that the SPR measurements throughout the manuscript were done using a single-cycle approach, where increasing concentrations of an analyte are injected onto the same ligand surface without surface regeneration and consequently no signal return to baseline. This is different than multi-cycle kinetic analysis which is probably what the Reviewer thought was done for our measurements. We clarify that now in three ways: 1) we mention in the Figure 2 legend that the SPR measurements are single-cycle; 2) we increased the size of the SPR panels in Figure 2c so that the individual injections can be seen more easily 3) we further emphasize this in the Methods section. Both single and multi-cycle kinetics are accurate ways to measure binding parameters and are commonly reported in literature (Kamat V and Rafique A, Analytical Biochemistry, 2017; Hossain et al, Cell Reports, 2022).

Signal convergence to equilibrium is not necessary for K_D determination using kinetics parameters and for a lot of binding interactions reaching equilibrium may not be feasible during the time course of the experiment. We respectfully disagree that the measurements are "noisy", as the measured curves are smooth and can be confidently fit to a 1:1 binding model as shown. In cases where the measurements reach equilibrium, as shown in Figure 2c top right panel (S2hlx-4 against DH1057 mAb) for example, the measurements at equilibrium have constant values across time, further highlighting that the signal is stable and consistent. We also note that in the revised manuscript we included multiple single-cycle measurements of CoV-2 spike binding to different antibodies (Figures S4, S8) and the values we report agree with those previously reported by other groups (Pinto et. al, Science, 2021; Gobeil et. al, Mol. Cell, 2022;), taking into account that we measured binding to the native SARS-CoV-2 spike without the 2P stabilization mutations, which shows tighter binding to S2 antibodies.

By looking at the curves in the edited Figure 2c, it is now clear that the slope of k_{on} from the measurement of S2hlx-7 binding to S2P6 mAb is much steeper/faster than that of S2hlx-4, which is consistent

with the measured values and with what the Reviewer was expecting to see. To further address the Reviewer's concerns, we share below additional measurements of S2hlx-4 and S2hlx-7 binding to S2P6 mAb using multi-cycle measurements. We note that for these data to be of publishable quality, additional optimization would likely be necessary to obtain even better curve fits. Nevertheless, these preliminary data illustrate that single and multi-cycle measurements are in agreement in terms of the measured kinetic rates and the overall K_D .

S2hlx-4/S2P6 mAb binding;

single cycle: $k_{on}=1.01 \times 10^4 \text{M}^{-1} \text{s}^{-1}$, $k_{off}=4.82 \times 10^{-4} \text{s}^{-1}$, $K_D=47.7 \text{nM}$;

multi-cycle: $k_{on}=5.4 \times 10^3 \text{M}^{-1} \text{s}^{-1}$, $k_{off}=5.564 \times 10^{-4} \text{s}^{-1}$, $K_D=103 \text{nM}$;

S2hlx-7/S2P6 binding;

single-cycle: $k_{on}=1.31 \times 10^7 \text{M}^{-1} \text{s}^{-1}$, $k_{off}=2.45 \times 10^{-3} \text{s}^{-1}$, $K_D=0.2 \text{nM}$;

multi-cycle: $k_{on}=2.58 \times 10^6 \text{M}^{-1} \text{s}^{-1}$, $k_{off}=9.05 \times 10^{-4} \text{s}^{-1}$, $K_D=0.35 \text{nM}$;

Minor issues

1. The most potent designs to bind CC40.8 are derived from S2hlx-Ex2, in which the epitope helix is an extension of the scaffold's terminal helix. The crystal structure of S2hlx-Ex19 showed that the epitope adopts a canonical alpha helix conformation without antibody binding. Since the designs derived from the other scaffolds are not as potent, it is questionable whether the designs are able to present the kinked helix conformation of the epitope. It would be better if the authors could discuss the limitation of the design method.

The Reviewer raises a relevant point and is correct in their assessment of S2hlx-Ex2. We do not think that displaying the epitope at the terminus is absolutely necessary for high antibody affinity. During the revision, we engineered and characterized a new epitope scaffold, S2hlx-54, now included in the manuscript, that is based on the topology of S2hlx-Ex3, where the grafted epitope is internal to the parent scaffold. While S2hlx-54 does not bind the three stem helix antibodies tested as tight as the S2hlx-Ex2 design, it nevertheless shows high affinities for them (K_D s of 45nM, 222nM, and 150nM against S2P6, DH1057.1, and CC40.8 mAb respectively). This suggests that alternative epitope presentation in other molecular contexts unexplored here may achieve high antibody binding. However, how to stabilize and present an epitope that needs to adopt two distinct conformations depending on the target mAb it engages is challenging. We discuss the limitations of our approach related to this and potential ways to overcome them in the Discussion (lines 471-492).

2. Some of the kinetic parameters in Fig. 1c and Fig. 2b are marked as "not tested." Although they probably don't affect the main conclusion of the paper, it would be better if fill these values in the main figures.

We performed additional measures and edited the figures such that no "not tested" values are shown in the main figures.

Reviewer #1 (Remarks to the Author):

The additional experiments and text provided by the authors have addressed my previous concerns. The manuscript is much approved and is suitable for publication.

Reviewer #2 (Remarks to the Author):

The authors have addressed the concerns raised by this reviewer.

Reviewer #3 (Remarks to the Author):

The authors addressed my comments. I agree to accept the manuscript in its current format.